# Perineuronal nets decrease membrane capacitance of peritumoral fast spiking interneurons in a model of epilepsy

Bhanu P. Tewari [1], Lata Chaunsali[1], Susan L. Campbell[1,2], Dipan C. Patel[1], Adam E. Goode[3] & Harald Sontheimer [1,4]

Brain tumor patients commonly present with epileptic seizures. We show that tumor-associated seizures are the consequence of impaired GABAergic inhibition due to an overall loss of peritumoral fast spiking interneurons (FSNs) concomitant with a significantly reduced firing rate of those that remain. The reduced firing is due to the degradation of perineuronal nets (PNNs) that surround FSNs. We show that PNNs decrease specific membrane capacitance of FSNs permitting them to fire action potentials at supra-physiological frequencies. Tumor-released proteolytic enzymes degrade PNNs, resulting in increased membrane capacitance, reduced firing, and hence decreased GABA release. These studies uncovered a hitherto unknown role of PNNs as an electrostatic insulator that reduces specific membrane capacitance, functionally akin to myelin sheaths around axons, thereby permitting FSNs to exceed physiological firing rates. Disruption of PNNs may similarly account for excitation-inhibition imbalances in other forms of epilepsy and PNN protection through proteolytic inhibition may provide therapeutic benefits.

[1] Glial Biology in Health, Disease, and Cancer Center, Virginia Tech Carilion Research Institute, 2 Riverside Cir., Roanoke, VA 24016, USA. [2] Department of Animal and Poultry Sciences, Virginia Tech, 3460 Litton Reaves Hall, Blacksburg, VA 24061, USA. [3] Virginia Tech Carilion School of Medicine and Research Institute, 2 Riverside Cir., Roanoke, VA 24016, USA. [4] School of Neuroscience, College of Science, Virginia Tech, 300 Turner Street NW, Blacksburg, VA 24061, USA. Correspondence and requests for materials should be addressed to H.S. (email: sontheim@vt.edu)

Seizures are common in patients with primary brain tumors and up to 80% of patients report at least one seizure prior to their diagnosis[1]. Seizures recur in about a third of these patients and give rise to tumor-associated epilepsy, often refractory to available antiepileptic drugs[2]. The cellular and molecular changes underlying tumor-associated epilepsy collectively point to a decrease in GABAergic and increase in glutamatergic transmission as the major contributors[3–6].

The enhanced glutamatergic drive is the result of glutamate (Glu) being assiduously released from gliomas via system Xc transporter (SXC) encoded by the *SLC7A11* gene that is upregulated in about 54% of glioma patients[4]. The SXC transporter is an obligated Glu-cystine antiporter that supplies cystine for the synthesis of the cellular antioxidant glutathione[7], hence making Glu release a byproduct of the cell's redox defense system[8]. By killing neurons through Glu excitotoxicity, gliomas use glutamate to create space for tumor expansion[9] and use it as an autocrine proinvasive signal[10]. Indeed, inhibition of SXC slows tumor growth[4,11] and reduces seizure frequency[6] in tumor-bearing mice. Consistent with these findings, a recent clinical study confirmed reduced Glu release in glioma patients acutely treated with the SXC inhibitor sulfasalazine[4]. Glu release, while necessary, is insufficient to drive peritumoral epilepsy[3]. Rather an additional loss of GABAergic inhibition is required with at least two suspected contributors; the peritumoral brain shows ~35% reduction in the density of fast spiking GABAergic interneurons[3] and the remaining GABAergic neurons show a significantly reduced inhibitory potential. The latter may be secondary to a change in the cell's chloride ($Cl^-$) equilibrium potential[3,5] rendering GABA currents less inhibitory.

A majority of cortical GABAergic neurons implicated in seizure disorders are parvalbumin-expressing fast spiking (100–800 Hz) interneurons ($PV^+$ FSNs) comprising 40–50% of all GABAergic neurons and are specialized in generating robust feed forward inhibition[12,13]. About 80% of $PV^+$ FSNs[14] are surrounded by perineuronal nets (PNNs) which are complex lattice-like extracellular matrix (ECM) assemblies of chondroitin sulfate proteoglycans (CSPGs), tenascin-R, hyaluronan and link proteins[15]. Functionally, PNNs are suggested to stabilize synaptic contacts[16] and possibly encode long-term memories[17]. PNNs may also restrict local ionic concentration due to a high density of their negatively charged constituent glycosaminoglycans (GAGs)[18], thereby regulating the microenvironment of FSNs and influencing their intrinsic properties[19]. Importantly, enzymatic digestion of GAGs alters the transmembrane $Cl^-$ gradient causing a depolarizing shift in the reversal potential of $GABA_AR$[20].

Interestingly, PNNs constituents are also substrates for matrix degrading enzymes including matrix metalloproteinases (MMPs), a disintegrin and metalloproteinase (ADAMs) and a disintegrin and metalloproteinase with thrombospondin motifs (ADAMTs), which are known to be released from gliomas[21], and hence may be subject to degradation by invading tumor cells.

The current study sought to gain a better understanding of how glioma-secreted molecules affect peritumoral neurons and the surrounding ECM to cause tumor-associated epilepsy. Using a clinically relevant glioma model, in which patient-derived xenolines were implanted into *scid* mice, we show that the peritumoral cortex (PTC) within 0.6 mm of tumor is plagued by excitotoxic neuronal cell death particularly FSNs. Furthermore, FSNs in the PTC show degraded PNNs caused by the proteolytic activity of MMPs released from the tumor. We show that PNN degradation alone is sufficient to reduce the firing frequency of FSNs thereby decreasing inhibitory tone, implicating PNN degradation as a major contributor to the loss of GABAergic inhibition in glioma-associated epilepsy. Importantly, real-time digestion of PNNs suggests that PNNs act as an insulator that decreases specific

membrane capacitance of the cell thereby permitting an increase in the maximally achievable firing rate of FSNs.

We describe a previously overlooked role of PNNs, namely to act as an insulator akin to the myelin sheath to reduce specific membrane capacitance of FSNs thereby permitting supraphysiological firing rates. These findings have profound implications that go well beyond tumor-associated epilepsy since it is likely the case for FSNs throughout the brain and may explain the changes in their firing properties in other diseases or following injury where MMPs are released in the context of inflammation and tissue reorganization.

## Results

**Neuronal loss and PNN disruption in the peritumoral brain.** We[6] and others[22] previously reported that glioma-released Glu contributes to peritumoral seizures presumably through chronic overactivation of neuronal Glu receptors enhancing the excitatory drive in the cortex. Subsequently, we also showed a concomitant decrease in the $PV^+$ interneurons[3]. Given the well-established Glu neurotoxicity for cortical neurons[23], we hypothesized that peritumoral loss of both $PV^+$ interneurons and principle neurons occur as a consequence of Glu release from the tumor.

To test this hypothesis, we implanted a highly epileptogenic patient-derived glioma xenoline (GBM22)[4] that avidly releases Glu via SXC into the cortex of *scid* mice and evaluated the PTC for evidence of neuronal loss using pan-specific neuronal marker, NeuN and two specific markers for FSNs—PV and Wisteria Floribunda Agglutinin (WFA). Since ~80% of $PV^+$ interneurons are surrounded by PNN, which is identified by WFA staining ($WFA^+$), and virtually all PNN-surrounded neurons in the cerebral cortex are FSNs[14,19], both PV and WFA were used as a marker to identify FSNs (Supplementary Fig. 1a, b). These three markers allowed us to quantitatively examine the overall neuronal loss as a function of distance from the implanted tumor (identified by high density of 4′,6-diamidino-2-phenylindole (DAPI)-labeled cell nuclei, Fig. 1a) and the relative change in inhibitory ($PV^+/WFA^+$) to all NeuN$^+$ neurons. NeuN$^+$ cells showed ~3-fold lower density in the immediate tumor vicinity than that in contralateral or sham-injected controls (Fig. 1a, b). The gradient of NeuN$^+$ cell loss was observed up to 0.6 mm from the tumor margin, the furthest distance evaluated, with significantly lower cell density compared to sham and contralateral brains (Fig. 1a).

In agreement with the literature[24–26], ~10% of all NeuN$^+$ neurons in cortex were $PV^+$ (Fig. 1b, c) and showed even greater distance-dependent decrease in density with >5-fold decrease in the closest proximity of tumor (Fig. 1c). While it is possible for PV expression to decrease without an actual loss of interneurons[27], this is not likely the case here for the following reasons: a majority of $PV^+$ interneurons are surrounded by PNN, which are identified by WFA staining, yet we saw a concurrent decrease in $WFA^+$ cells (Fig. 1d), and did not see a decrease either in the ratio of $PV^+/WFA^+$ or in the population of NeuN$^+WFA^+PV^-$ cell types (Supplementary Fig. 1c) between tumor-implanted and sham-injected mice as would be expected if cells survived yet only their PV expression was altered, leading us to conclude that $PV^+$ interneurons are significantly decreased in number in the PTC.

To investigate whether inhibitory neurons are preferentially affected by the tumor, we compared relative densities of different cell types independent of their overall abundance. We normalized the cell density to that observed in sham animals and plotted them directly comparing NeuN$^+$, $PV^+$, and $WFA^+$ neurons (Fig. 1e). These data suggest that while the cell density of all PTC neurons was gradually decreased as approaching the tumor,

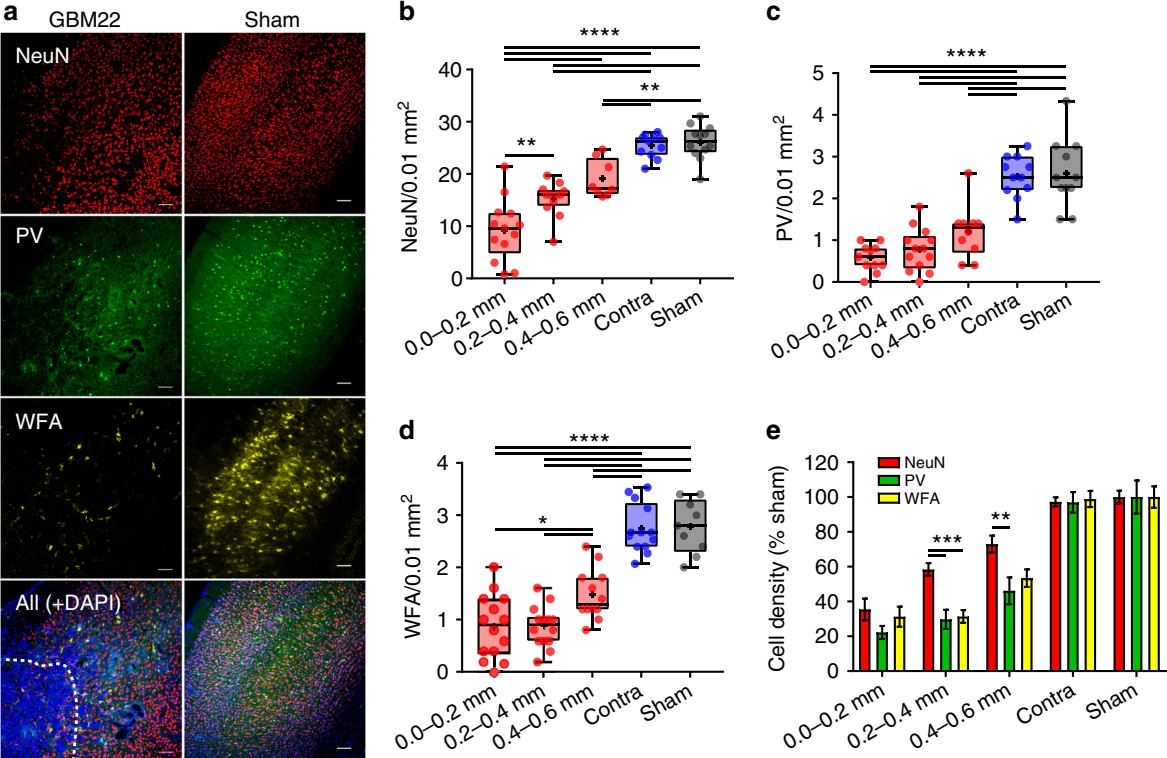

**Fig. 1** Glioma-mediated neuronal cell loss with a preferential vulnerability of PV interneurons. **a** Representative images of NeuN (red), PV (green), and WFA (yellow) labeling, top to bottom, in glioma-injected (left) and sham-injected (right) cortical slices. Glioma mass (bottom left) can be discerned by highly dense DAPI-positive cell nuclei delimited by dashed line. Scale = 100 μm. **b** Mean density of NeuN+ neurons at 0.0–0.2 mm (9.25 ± 1.63), 0.2–0.4 mm (15.25 ± 0.94), and 0.4–0.6 mm (19.06 ± 1.28) distance from the glioma mass compared to equivalent position in contralateral (25.41 ± 0.68) and sham animals (26.12 ± 0.94). $n = 6$ mice. **c** Summary of PV+ neuron cell density at 0.0–0.2 mm (0.58 ± 0.1), 0.2–0.4 mm (0.77 ± 0.14), and 0.4–0.6 mm (1.2 ± 0.2) distance from the glioma mass compared to contralateral (2.52 ± 0.15) and sham (2.6 ± 0.25). $n = 6$ mice. **d** WFA+ PNNs in the PTC at 0.0–0.2 mm (0.87 ± 0.16), 0.2–0.4 mm (0.87 ± 0.1), and 0.4–0.6 mm (1.48 ± 0.14), from glioma mass compared to contralateral (2.75 ± 0.13) and sham (2.78 ± 0.17). $n = 6$ mice. **e** Normalized density of PV+ and NeuN+ neurons at 0.2–0.4 mm (PV+, 29.75 ± 5.45%; NeuN+, 58.39 ± 3.62%), and 0.4–0.6 mm (PV+, 46.18 ± 7.7%; NeuN+, 72.99 ± 4.88%) distance from the tumor border. No significant difference was observed in the density of PV+ and NeuN+ neurons in contralateral (PV+, 96.91 ± 5.90%; NeuN+, 97.29 ± 2.6%) and sham-injected animals (PV+, 100 ± 9.58%; NeuN+, 100 ± 3.62%). Normalized density of WFA+ PNNs in the 0.2–0.4 mm (PNN+, 31.37 ± 3.66%; NeuN+, 58.38 ± 3.61%) was significantly different compared to NeuN. $n = 6$ mice. Cell density was normalized to sham. Bar data represent mean ± SEM unless otherwise stated. ****$P < 0.0001$, ***$P < 0.001$, **$P < 0.01$, *$P < 0.05$, one-way ANOVA, Tukey's post-hoc test in (**b**−**d**); and two-way ANOVA, Tukey's post-hoc test in (**e**)

WFA+/PV+ interneurons was significantly more affected, particularly within 0.2–0.6 mm PTC.

The gradient of PNN degradation was also observed in the PTC as intact PNNs were only observed at >0.4 mm PTC, while in closer proximity, the PNNs appeared damaged, fragmented, and often limited to the cell body and no longer containing a cell (Fig. 2a, Supplementary Fig. 1d). A PNN degradation gradient in the PTC was further evident by graded WFA intensities in different peritumoral spatial bins (Fig. 2b), which also suggest disintegration of lattice-like assembly of PNNs. We confirmed the same by observing significant reduction in the WFA intensity peaks (see methods) in the line profile of peritumoral PNNs, an indicative of large holes in the PNN (Fig. 2c−e).

To question the role of glutamate in peritumoral neuronal death, we repeated the above experiments with a patient-derived glioma (GBM14) that does not express the SXC transporter and consequently does not release Glu[4]. Consistent with Glu being essential for peritumoral neurotoxicity, GBM14 PTC showed no significant loss of PV+ interneurons in the >0.2 mm PTC (Fig. 3a, b), whereas the Glu-releasing tumors (GBM22) decreased NeuN+ cells to ~50–60% within 0.2–0.6 mm PTC. Only in the immediate PTC (0.0–0.2 mm) did both tumor types exhibit identical neuronal cell loss. In GBM14, WFA staining showed largely

intact PNNs even in close proximity to the tumor (0.2–0.4 mm), whereas in GBM22, PNNs were largely degraded (Fig. 3a, c).

To question whether PNN degradation by tumor is specific for PV+ interneurons, we implanted GBM22 tumors near the hippocampal CA2 region where dense CSPGs surround excitatory neurons similar to the cortical PNNs (Fig. 3d). The interstitial matrix of the *stratum radiatum* near CA2 also showed dense CSPGs (Fig. 3d). Here too, tumor degraded dense CSPGs in the *statrum pyramidale* (Fig. 3d, e, compare squares) and *statrum radiatum* (Fig. 3d, f, compare circles) suggesting that gliomas degrade PNNs without any predilection for PV+ interneurons.

Taken together, these data suggest that tumor-secreted molecules, likely Glu and proteolytic enzymes, cause a distance-dependent loss of interneurons and degradation of PNNs, respectively.

**Glioma-released proteases cause PNN degradation**. Gliomas release several ECM remodeling/degrading enzymes[28] including MMPs[29], for which PNN proteoglycans are known substrates[21], and enhanced MMPs activity is also prominent in the CNS under pathological conditions[30] that present with astrogliosis. Given the presence of reactive astrogliosis in the PTC (Fig. 4a, b), we hypothesized that reactive astrogliosis may contribute to

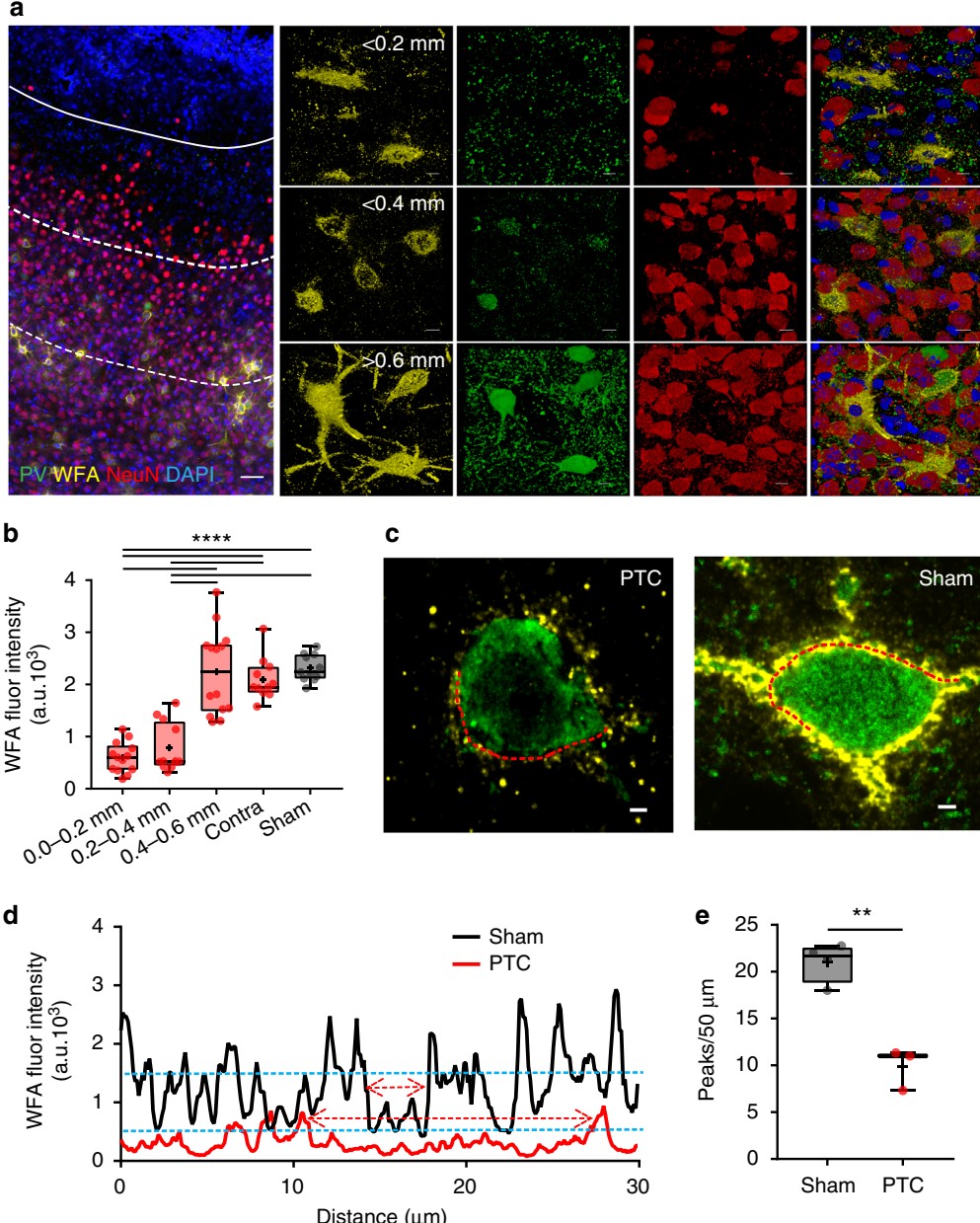

**Fig. 2** Disintegration of PNNs in the peritumoral cortex. **a** Confocal images of GBM22 PTC labeled with DAPI (blue), WFA (yellow), PV (green), and NeuN (red) showing spatial gradient of alterations. The distance between tumor border (white solid line) and each of the dotted lines is ~0.2 mm. Top panel images from <0.2 mm PTC show disintegrated PNNs (yellow) mostly lacking PV and NeuN and reduced neuronal cell numbers in close proximity of tumor. Middle panel images, in the 0.2–0.4 mm PTC, show a much higher number of NeuN+ neurons but few that contain PV. Bottom panel images from >0.6 mm PTC show mostly intact PNNs around PV neurons and high neuronal cell density (scale = 50 μm and 5 μm in large image and panel images, respectively) which are indistinguishable from sham (Supplementary Figure 1a-b). **b** WFA intensity within 0.0–0.2 mm (599.8 ± 80.84 au) and 0.2–0.4 mm (776.7 ± 134.8 au) PTC was significantly lower than sham (2315.0 ± 76.28 au) and contralateral (2012 ± 121.6 au). $n = 6$ mice. ****$P < 0.0001$, one-way ANOVA, Tukey's post-hoc test. **c** Confocal images of two representative PV neurons showing disintegrated architecture in GBM22 PTC compared to sham. Scale = 2 μm. **d** WFA fluorescence intensity of a line drawn along the periphery of PV neuron in the PTC (left in **c**) and sham (right in **c**) showing many high-intensity WFA peaks in sham compared to the PTC. Upper and lower blue dotted lines represent the threshold WFA intensity (50% of the highest intensity) for the PTC and sham, respectively. Upper and lower red two-headed arrows within two nearest WFA peaks indicate the size of a hole in PNN from the PTC and sham, respectively. **e** Disintegrated PNNs in the PTC show significantly lower numbers of WFA intensity peaks (sham, 21.15 ± 0.92, $n = 13$ PNNs from four mice; PTC, 9.88 ± 0.84, $n = 9$ PNNs from three mice). **$p < 0.01$, unpaired $t$ test. PTC in (**d**) and (**e**) represent 0–0.4 mm area

peritumoral PNN degradation. However, three lines of reasoning suggest otherwise. Firstly, while the PTC in close proximity to the tumor (<0.3 mm) showed disintegrated PNNs surrounded by highly reactive astrocytes (Fig. 4a (bottom square), Fig. 4b, c), we observed essentially intact PNNs around equally high GFAP (glial fibrillary acidic protein) expressing astrocytes in more distal (>0.6 mm) PTC (Fig. 4a (top square), 4b, c). Secondly, PNNs are intact in an astrocyte-specific beta integrin knockout (β1$^{-/-}$) mouse (Fig. 4d−g, Supplementary Fig. 1e), which is a genetic model of astrogliosis and spontaneous seizures[31]. Thirdly, peritumoral reactive astrocytes did not show detectable MMPs/gelatinase activity (Supplementary Fig. 2a).

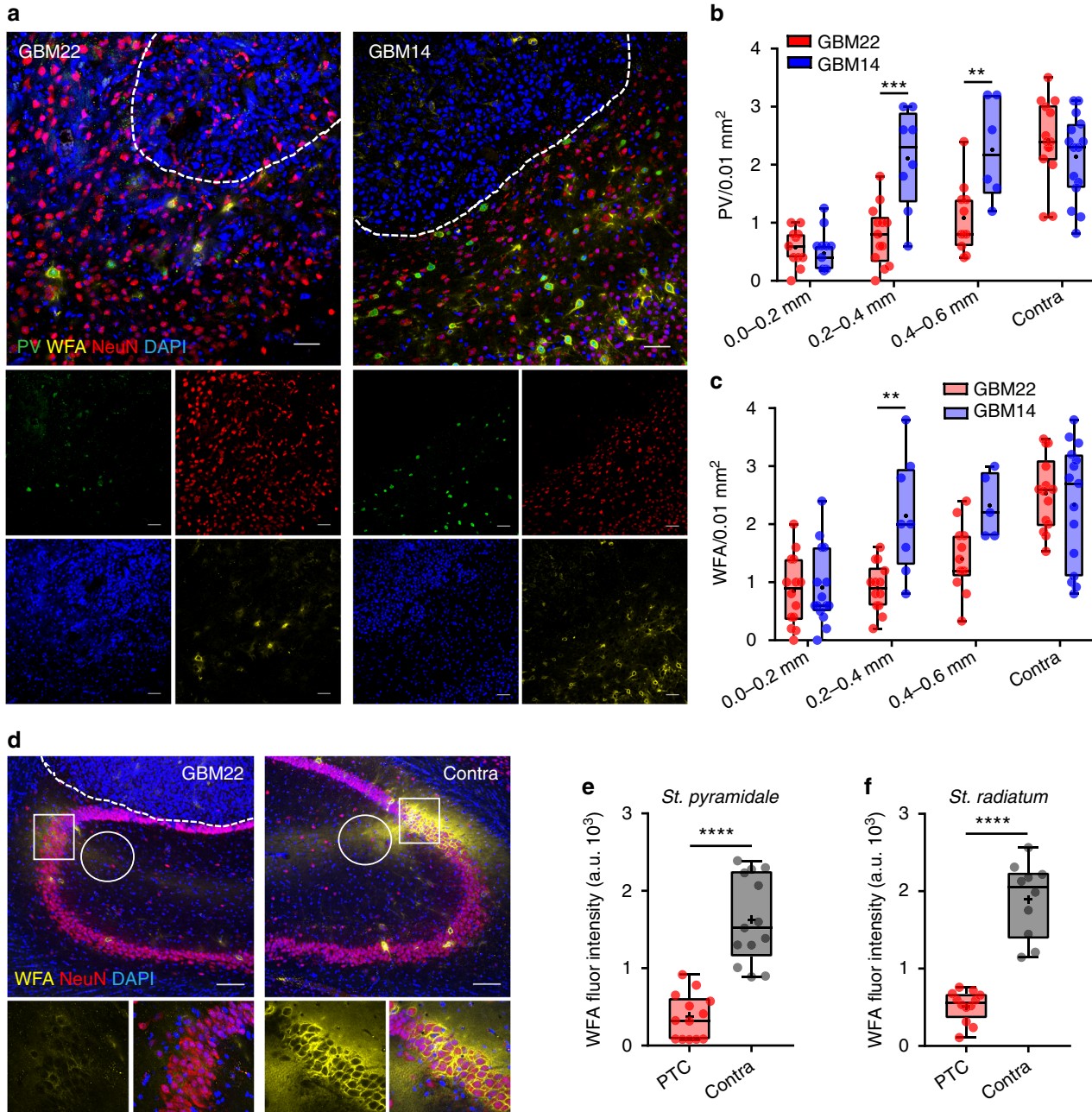

**Fig. 3** Differential effects of epileptogenic GBM22 and nonepileptogenic GBM14 gliomas on cortical PV⁺ neurons and PNNs in the PTC. **a** Representative confocal images of DAPI (blue), PV (green), PNN (yellow), and NeuN (red) immunofluorescence in the PTC of GBM22 (left) and GBM14 (right). Scale = 20 μm. **b** Peritumoral PV⁺ cells density in GBM22 and GBM14 slices at 0.0–0.2 mm (GBM22, 0.58 ± 0.1; GBM14, 0.48 ± 0.1), 0.2–0.4 mm (GBM22, 0.79 ± 0.14; GBM14, 2.1 ± 0.31), and 0.4–0.6 mm (GBM22, 1.08 ± 0.18; GBM14, 2.26 ± 0.35) distance from the tumor ($n = 6$ mice in each bin). **c** The density of PNN⁺ cells in the PTC of GBM22 and GBM14 slices at 0.0–0.2 mm (GBM22, 0.85 ± 0.16; GBM14, 0.91 ± 0.17), 0.2–0.4 mm (GBM22, 0.9 ± 0.11; GBM14, 2.15 ± 0.35), and 0.4–0.6 mm (GBM22, 1.40 ± 0.16; GBM14, 2.32 ± 0.25). $n = 6$ mice in each bin. The density of PV⁺ neurons (**b**), and PNNs (WFA) (**c**) beyond 0.6 mm PTC in GBM14 and GBM22 was not significantly different from contralateral and sham. $n = 6$ mice. **d** Representative confocal images of DAPI (blue), PV⁺ (green), PNN⁺ (yellow), and NeuN (red) immunofluorescence from GBM22-implanted ipsilateral and contralateral hippocampal slices. Scale = 100 μm top and 20 μm bottom images. **e** WFA intensity in CA2 PNNs (peritumoral, 376.8 ± 81.12; contralateral, 1628 ± 155.2) $n = 4$ mice. **f** WFA intensity in stratum radiatum (peritumoral, 517.7 ± 57.66; contralateral, 1894 ± 153.2) $n = 4$ mice. All fluorescence intensities represent arbitrary units of fluorescence. ****$P < 0.0001$,***$P < 0.001$, **$P < 0.01$, two-way ANOVA, Tukey's post-hoc test in (**b**) and (**c**); Welch's $t$ test in (**e**) and (**f**)

MMP-2 and MMP-9, also known as gelatinases, are correlated with increasing malignancy in high-grade glioblastoma[32]; hence, we examined their potential involvement in peritumoral PNN degradation. We performed in situ zymography (ISZ) using gelatinase assays in frozen brain slices to evaluate the activity of MMP-2 and MMP-9 (DQG/gelatinase). Our data show higher gelatinase activity in glioma-injected than in sham-treated brain slices (Fig. 5a, b). Further, gelatinase activity was significantly higher in glioma cells than in peritumoral cortical cells. Basal gelatinase activity in sham and contralateral hemisphere was

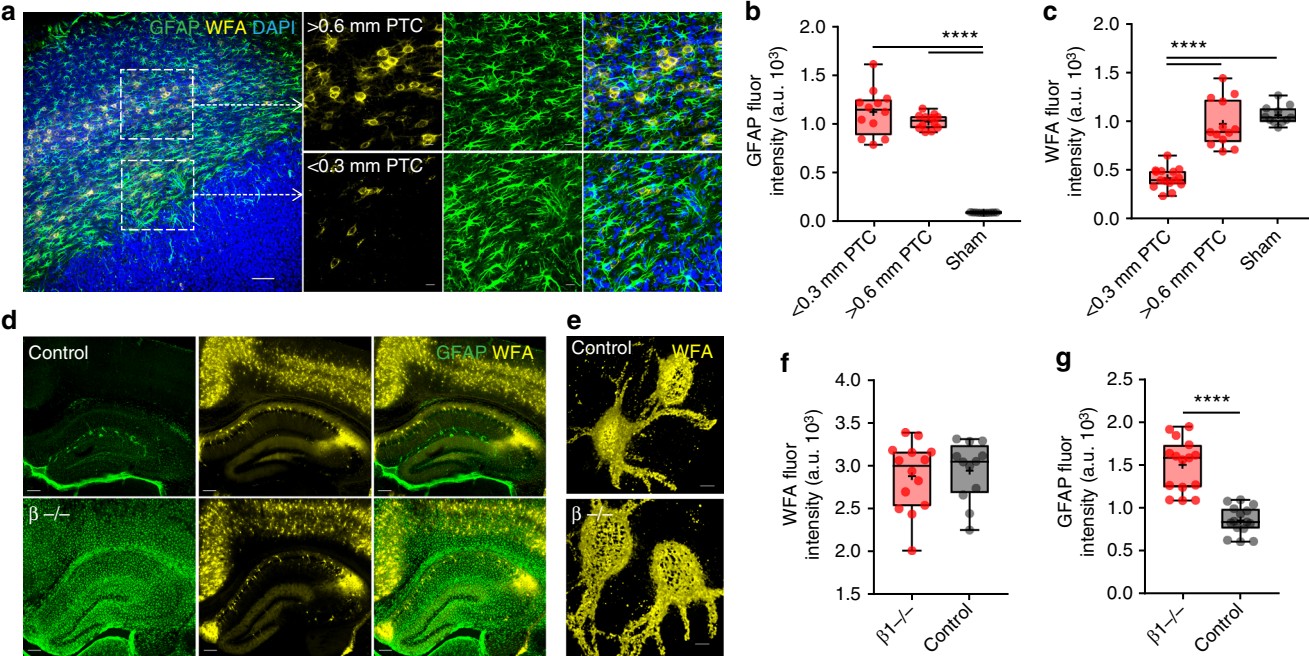

**Fig. 4** PNN disintegration is not correlated with astrogliosis. **a** Confocal images of GBM22 PTC labeled with GFAP (green), WFA (yellow), and DAPI (blue) showing reactive astrocytes near disintegrated PNNs (bottom square and corresponding lower panels) and intact PNNs (upper square and corresponding upper panels) in proximal (0.0–0.3 mm) and distal (>0.6 mm) PTCs, respectively. Scale = 100 μm in the main image and 20 μm in the magnified images. Sham-treated controls did not show any reactive astrogliosis (Supplementary Fig. 1e). **b** High GFAP expression in the PTC (1127 ± 68.23, $n = 5$ mice) compared to sham (97.51 ± 0.83, $n = 6$ mice), but similar between proximal and distal PTCs (proximal PTC, 1127 ± 68.23; distal PTC, 1031 ± 22.08; $n = 5$ mice). **c** WFA intensity was significantly lower in proximal PTC (0.0–0.3 mm, 418.9 ± 27.06) compared to both distal PTC (>0.6 mm, 970.6 ± 66, $n = 5$ mice) and sham (1063 ± 28.47, $n = 6$ mice). **d** Representative images of GFAP and WFA immunofluorescence in wild-type (control) (top) and $β1^{-/-}$ (bottom) cortical sections showing reactive astrogliosis in the latter and intact PNNs. Scale = 200 μm. **e** Representative high-magnification 3D volume projection images of individual PNNs showing comparable WFA intensity and PNN architecture in wild-type (control) and $β1^{-/-}$. Scale = 5 μm. **f** Box and whisker plot showing comparable WFA intensity in wild-type (control) (2942 ± 98.72) and $β1^{-/-}$ (2881 ± 105.3). $n = 8$ mice. **g** Significantly higher GFAP expression in $β1^{-/-}$ (1500 ± 77.83) than wild-type (control) (856.7 ± 42.13). $n = 8$ mice. ****$P < 0.0001$, one-way ANOVA, Tukey's post-hoc test in (**b**) and (**c**); unpaired $t$ test in (**f**); and Welch's $t$ test in (**g**). All fluorescence intensities represent arbitrary units of fluorescence

significantly lower than that in the PTC and glioma mass (Fig. 5b, Supplementary Fig. 2b).

PTC exhibited a spatial gradient of higher to lower gelatinase activity from the glioma border to the distal PTC (Fig. 5b) that appeared negatively correlated with the WFA intensity (Fig. 5c), suggesting the tumor as the likely source of MMPs. Corroborating our IHC data, PNN disintegration was evident by significantly lower WFA intensity within ~0.6 mm PTC (Fig. 5c). PTC also exhibited reactive astrogliosis; however, an absence of gelatinase activity in reactive astrocytes ruled them out as the primary source of MMPs (Supplementary Fig. 2a).

If glioma-released MMPs and/or glioma-induced MMPs release from the CNS cells causes PNN degradation, then blocking MMPs should rescue PNNs in glioma-injected brains. To examine this, we treated GBM22-implanted mice with GM6001 (Illomastat, 100 mg/kg, i.p.), a broad-spectrum inhibitor of MMPs. Confirming our prediction, GM6001 treatment reduced gelatinase activity within the glioma (Fig. 5a, middle panels) and in the PTC (Fig. 5d). GM6001-treated mice showed a higher number of PNNs in the PTC (Fig. 5e) that may be a consequence of the preservation of PNNs which are known to protect cells from excitotoxicity[33]. Indeed, GM6001-treated mice retained structural integrity of PNNs in the PTC (Fig. 5f–i) as evident by the presence of PNNs around cell body and proximal dendrites (Fig. 5f, middle panels), more WFA peaks (Fig. 5g, h), and higher WFA intensity (Fig. 5i) compared to sham-treated mice. As PNNs protect FSNs from oxidative stress[33], we suggest that preventing PNN degradation potentially may also protect the

cells from glutamate excitotoxicity-induced oxidative stress. These data suggest that the presence of glioma induces increased MMPs activity that degrades peritumoral PNNs.

**Biophysical changes in peritumoral neurons.** Seizures originate from the PTC, and previous studies[3,5,6] show altered excitation−inhibition balance in the PTC due to changes in glutamatergic and GABAergic activity. The latter activity originates from the FSNs that express PNNs, which as we show above, are degraded in the PTC. Since PNNs are presumed to influence the ionic microenvironment around the FSNs[18,19] and stabilizes synaptic contacts[34], it is plausible that PNN degradation by glioma-released proteases may alter the physiological properties of FSNs. To address this question, we obtained patch-clamp recordings from neurons within ~600 μm of the PTC, an area most affected by the tumor (Fig. 1a, e). We used characteristic discharge patterns to distinguish between FSNs (>100 Hz) with little or no spike adaptation[35] and slower firing (15–40 Hz) excitatory neurons (Supplementary Fig. 1f). Recording patch pipettes also contained the green fluorescent tracer, Lucifer yellow (LY), allowing us to identify cells postrecording to determine their distance from the tumor and PNN morphology on WFA staining (Supplementary Fig. 3a, b). These biophysical characteristics and postrecording WFA staining allowed us to unequivocally determine the specific type of the recorded cells.

Peritumoral excitatory neurons often showed spontaneous epileptiform discharges (Supplementary Fig. 4a), rarely observed

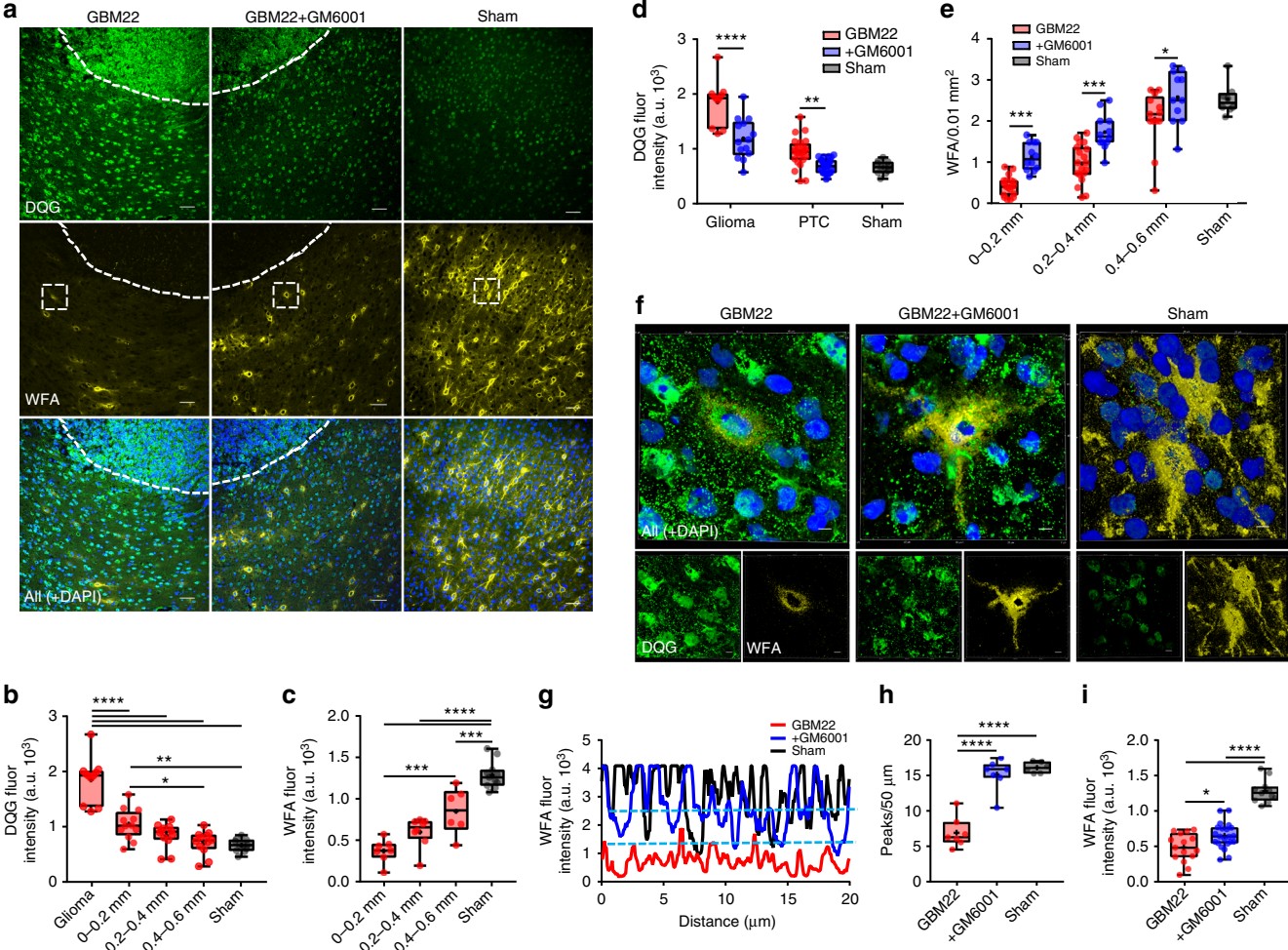

**Fig. 5** PNN degradation by glioma-released proteases. **a** Representative images showing gelatinase activity (DQG-green), DAPI (blue), and PNN (WFA-yellow) immunofluorescence in GBM22 PTC (left), GM6001-treated glioma-implanted PTC (GBM22 + GM6001, middle), and sham (right) cortical sections. Scale = 50 μm. **b** Mean gelatinase activity in glioma mass (delimited by dashed line in (**a**)) (1851.47 ± 120.17), 0–0.2 mm (1048.74 ± 79.63), 0.2–0.4 mm (848.61 ± 64.25), and 0.4–0.6 mm (715.96 ± 61.89) PTC and sham (669.96 ± 34.56). $n = 6$ mice. **c** WFA fluorescent intensity at 0–0.2 mm (372.65 ± 54.10), 0.2–0.4 mm (599.17 ± 64.94), 0.4–0.6 mm (849.07 ± 113.94) PTC and sham (1282.73 ± 49.03). $n = 5$ mice. **d** Mean gelatinase activity in glioma mass (GBM22, 1851.47 ± 120.17; +GM6001, 1189.96 ± 98.43) and PTC (GBM22, 948.68 ± 54.21; +GM6001, 692.80 ± 23.61) was significantly lower in GBM22 + GM6001 group (also referred as +GM6001) than the GBM22 (nontreated) counterparts. $n = 5$ mice. **e** PNN numbers in +GM6001 group were significantly higher in 0.0–0.2 mm (GBM22, 0.43 ± 0.05; +GM6001, 1.14 ± 0.09), 0.2–0.4 mm (GBM22, 0.98 ± 0.10; +GM6001, 1.73 ± 0.12) and 0.4–0.6 mm PTC (GBM22, 2.05 ± 0.19; +GM6001 2.58 ± 0.19). $n = 5$ mice. **f** Representative 3D volume projections from within 0–0.4 mm PTC (dotted square in (**a**)) showing gelatinase activity (DQG) and PNN architecture (WFA) in GBM22, +GM6001 and sham groups. Scale = 5 μm. **g** Line intensity profile of a single PNN (in **f**) shows high intensity WFA peaks in GBM22, +GM6001 and sham groups. Dashed lines represent the threshold (50% of max WFA intensity) for WFA peak counts (in **h**). **h** Mean WFA peaks in PNNs from GBM22 (7.08 ± 0.63), +GM6001 (15.16 ± 0.65) and sham (16.05 ± 0.53). Data of GBM22, +GM6001, and sham are from 20, 22, and 17 PNNs, respectively ($n = 4$ mice). **i** Mean WFA fluorescence intensity in +GM6001 PTC was higher than GBM22 PTC (+GM6001, 656.96 ± 37.77; GBM22, 493.46 ± 51.21). $n = 6$ mice. PTC data in (**d**), (**h**), and (**i**) are from within 0–0.4 mm of tumor border. ****$P < 0.0001$, ***$P < 0.001$, **$P < 0.01$, *$P < 0.05$, one-way ANOVA, Tukey's post-hoc test in (**b, c, h, i**); two-way ANOVA, Sidak's post-hoc test in (**d, e**). All fluorescence intensities represent arbitrary units of fluorescence

in contralateral and shams. PTC neurons were also significantly depolarized (Fig. 6a); however, their membrane capacitance (Cm) and input resistance ($R_{in}$) remained unaffected (Fig. 6b, c). They also showed lower firing threshold i.e., minimum current required to elicit an action potential (Fig. 6d) and had a significantly higher firing frequency (input−output curve) (Fig. 6e, f). To test the hypothesis that lower firing threshold and the depolarized Vm of the PTC neurons were due to GBM22-released Glu, we studied the above properties in the PTC of GBM14-implanted mice. This indeed nullified the changes in firing threshold and Vm with no apparent difference to sham or contralateral brains (Fig. 6e, Supplementary Fig. 4b, c). In

addition, the spike frequency of excitatory neurons in GBM22 PTC was significantly reduced on blocking the glutamatergic neurotransmission, further confirming them driven by glioma-released Glu rather than intrinsic to the cell (Supplementary Fig. 5b).

Peritumoral FSNs were similarly depolarized (Fig. 6g) but showed significantly higher Cm (Fig. 6h). In contrast to excitatory cells, neither resting membrane potential nor input−output curve (Fig. 6j−l) could be attributed to the tumor-released Glu as they were identical in neurons with disintegrated PNNs recorded in the PTCs of GBM14 and GBM22-implanted brains (Fig. 6k). Spike frequency in GBM22 contralateral half remained similar to sham

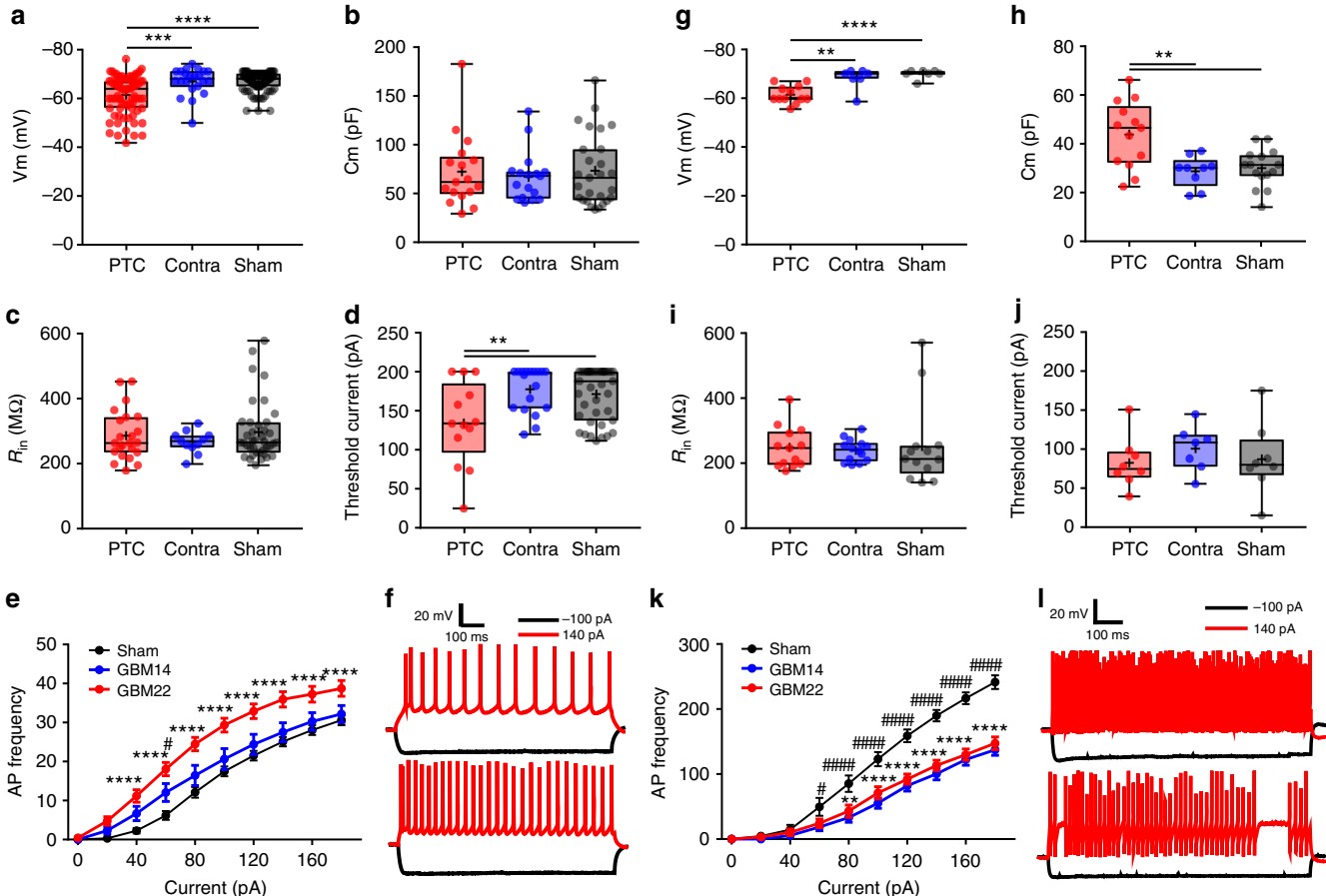

**Fig. 6** Altered biophysical properties of excitatory and inhibitory neurons in the PTC. **a** Depolarized excitatory neurons in the PTC (61.49 ± 0.86, $n = 80$ (11)) than in contralateral (67 ± 1.16, $n = 22$(6)), and sham (67.02 ± 0.38, $n = 98$(15)). **b** Membrane capacitance (Cm) of excitatory neurons (PTC, 72.90 ± 8.92, $n = 17$(10); contralateral, 67.46 ± 6.11, $n = 17$(8); sham, 73.93 ± 6.95, $n = 27$(10)). **c** Input resistance ($R_{in}$) of excitatory neuron in the PTC (PTC, 288.20 ± 15.82, $n = 23$(10); contralateral, 269.22 ± 10.35, $n = 11$(5); sham, 298.63 ± 14.47, $n = 39$(10)). **d** Threshold current of excitatory neurons in the PTC (134.76 ± 14.69, $n = 13$(8)); contralateral (177.55 ± 6.69, $n = 18$(8)); sham (171.33 ± 5.28, $n = 39$(10)). **e** Higher AP frequency of excitatory neurons in GBM22 PTC ($n = 27$(10)) than GBM14 ($n = 12$(7)), and sham ($n = 36$(10)). **f** Representative traces of excitatory neuron's firing from sham (upper) and GBM22 PTC (lower) on −100pA (black) and 140pA (red) current steps. **g** Depolarized Vm of FSNs in GBM22 PTC (61.67 ± 1.01, $n = 12$(8)) compared to contralateral (68.14 ± 1.59, $n = 7$(6)) and sham (69.85 ± 0.67, $n = 7$(6)). **h** Higher Cm of FSNs in GBM22 PTC (43.84 ± 3.74, $n = 13$(8)) than in contralateral (28.82 ± 2.14, $n = 9$(8)) and sham (30.11 ± 1.99, $n = 15$(6)). **i** $R_{in}$ of FSNs in the PTC (249.10 ± 17.65, $n = 13$(9)), contralateral (240.08 ± 9.07, $n = 14$(7)), and sham (252.64 ± 35.30, $n = 13$(7)) was not different. **j** Threshold current of FSNs in the PTC (82.75 ± 11.49, $n = 8$(6)), contralateral (100.57 ± 10.94, $n = 7$(5)) and sham (87.25 ± 16.06, $n = 8$(5)) was not different. **k** AP (Spike) frequency of FSNs in GBM22 PTC ($n = 18$(11)) and GBM14 PTC ($n = 19$(7)) was significantly lower than in sham ($n = 10$(7)). **l** Representative traces of FSN's response from sham (upper) and GBM22 PTC (lower) on −100pA (black) and 140pA (red) current steps. $n =$ cells(mice) in all. The units for membrane potential (Vm), threshold current, membrane capacitance (Cm) and input resistance ($R_{in}$) are mV, pA, pF, and MΩ, respectively. ****$P < 0.0001$, ***$P < 0.001$, **$P < 0.01$, one-way ANOVA in (**a**–**d**, **g**–**j**); two-way ANOVA in (**e**) (sham vs. GBM22) and (**k**) (GBM22 vs. sham); Tukey's post-hoc test in all. ####$P < 0.0001$, #$P < 0.05$, two-way ANOVA, Tukey's post-hoc test in (**e**) (GBM14 vs. sham) and (**k**) (GBM14 vs. sham)

(Supplementary Fig. 4e). In essence, electrophysiological attributes of FSNs with disintegrated PNNs (Supplementary Fig. 3c) in the PTC of GBM22 and GBM14 (Fig. 6g−l and Supplementary Fig. 4d) were identical. Most importantly, the firing frequencies of FSNs in both GBM22 and GBM14 PTCs were reduced by ~50% compared to sham (Fig. 6k). This was not due to altered Vm of FSNs in GBM22 PTC as clamping them at −60, −65, or −70 mV did not change their firing frequency (Supplementary Fig. 5a). Similarly, firing frequency of FSNs in sham remained unaltered at −60, −65, or −70 mV (Supplementary Fig. 5a). Also, the highest spike frequency sustained by FSNs was not significantly different at any holding Vm tested (Supplementary Fig. 5a right). In addition, there was no correlation between lowered firing frequency of FSNs and glutamate-releasing capability of GBM14 and GBM22. This was substantiated by observing no change in firing frequency of

FSNs in GBM22 PTC after blocking glutamatergic neurotransmission by APV and CNQX (Supplementary Fig. 5c). Therefore, FSNs in the PTC are intrinsically less excitable than their counterparts in the sham.

Since PNN degradation was correlated with altered FSN properties, we predicted that blocking PNN degradation due to glioma-released MMPs by GM6001 treatment should also retain the biophysical properties of enclosed FSN. GM6001 treatment did not alter $R_{in}$ or threshold current but the resting Vm (Fig. 7a, c, d). Most importantly, FSNs in GM6001-treated mice brain slices showed profound differences in Cm and input−output curves, and both were essentially identical to the sham (Fig. 7b, e, f). The recorded FSNs also showed largely intact PNNs in the PTC of GM6001-treated group compared to the nontreated group (Supplementary Fig. 3c).

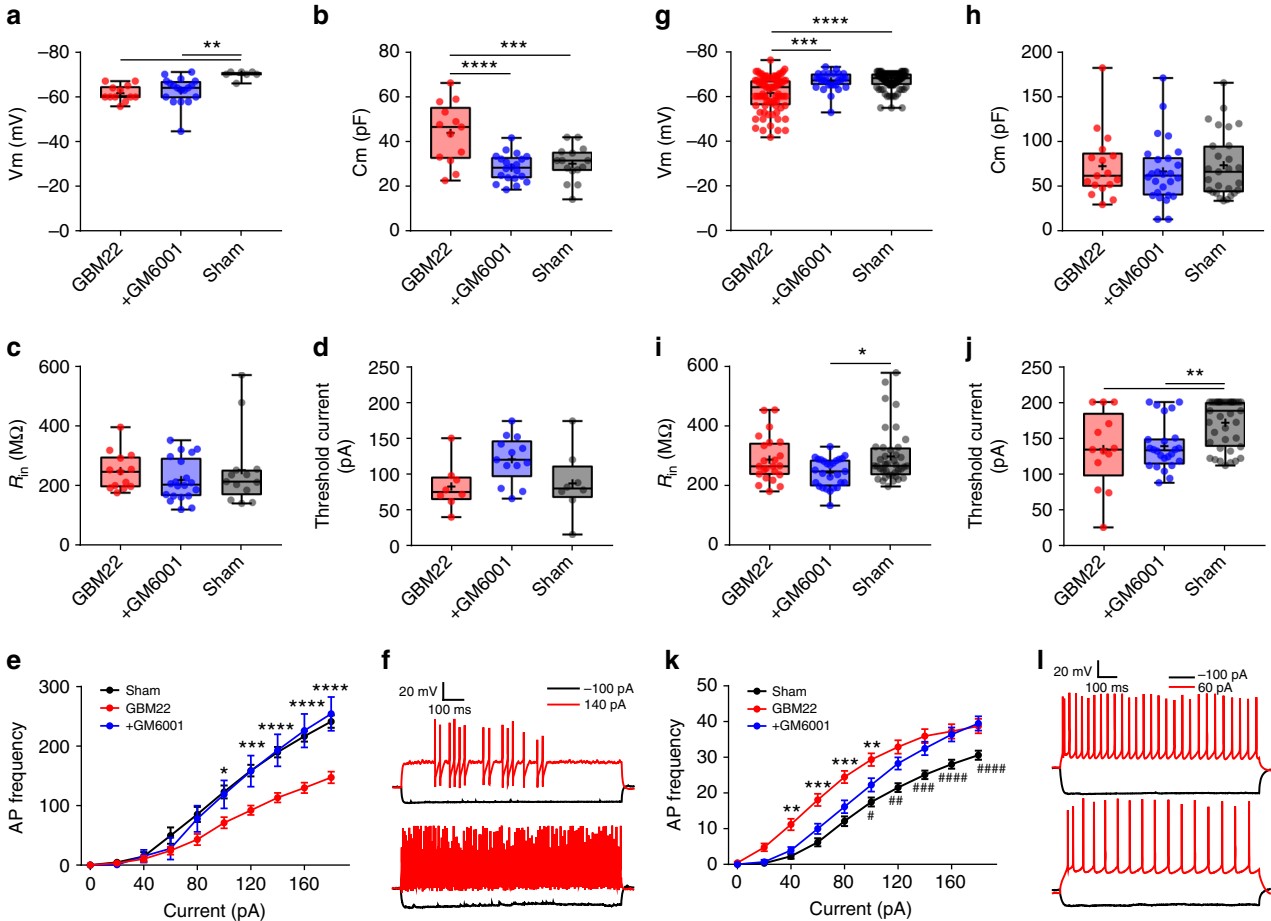

**Fig. 7** Electrophysiological properties of inhibitory (FSNs) and excitatory neurons in the PTC of GM6001-treated mice. **a** Unaltered resting membrane potential (Vm) of FSNs in the PTC of GM6001-treated (62.88 ± 1.40, $n = 18(5)$) than in nontreated (GBM22) PTC (61.67 ± 1.01, $n = 12(8)$), but significantly different than in sham (69.85 ± 0.67, $n = 7(6)$). **b** Lower Cm of FSNs in the PTC of GM6001-treated (28.27 ± 1.36, $n = 20(5)$) than nontreated (43.84 ± 3.74, $n = 13(8)$), but similar to sham (30.11 ± 1.99, $n = 15(6)$). **c** Unaltered $R_{in}$ of FSNs in GM6001-treated PTC (219.90 ± 15.69, $n = 20(5)$) compared to GBM22 PTC (249.10 ± 17.65, $n = 13(9)$) and sham (252.64 ± 35.30, $n = 13(7)$). **d** Threshold current of FSNs remained unaltered in GM6001-treated PTC (120.76 ± 8.96, $n = 13(5)$) compared to nontreated PTC (82.75 ± 11.49, $n = 8(6)$) and sham (87.25 ± 16.06, $n = 8(5)$). **e** Higher AP frequency of FSNs in GM6001-treated PTC ($n = 12(5)$) than in nontreated PTC ($n = 18(11)$) but not than in sham ($n = 10(7)$). **f** Representative Vm traces from nontreated (upper) and GM6001-treated PTC (lower) FSNs on −100pA (black) and 140pA (red) current steps. **g** Resting membrane potential (Vm) of excitatory neurons in GM6001-treated PTC (66.62 ± 0.77, $n = 27(8)$) remained similar to sham (67.02 ± 0.38, $n = 98(15)$), but was significantly higher than the nontreated PTC (61.49 ± 0.86, $n = 80(11)$). **h** Cm of excitatory neurons in GM6001-treated PTC remained unaltered (PTC, 72.90 ± 8.92, $n = 17$ (10); GM6001-treated PTC, 67.00 ± 6.95, $n = 26(8)$; sham, 73.93 ± 6.95, $n = 27(10)$). **i** Input resistance ($R_{in}$) of excitatory neurons in GM6001-treated PTC (244.64 ± 9.24, $n = 27(6)$) was not significantly different than in nontreated PTC (288.18 ± 15.82, $n = 23(10)$), but than in sham (298.63 ± 14.47, $n = 39(10)$). **j** Lower threshold current in excitatory neurons in nontreated PTC (134.76 ± 14.69, $n = 13(8)$) than in GM6001-treated PTC (138.75 ± 6.81, $n = 24$ (8)) and sham (171.33 ± 5.28, $n = 39(10)$). **k** Higher AP (firing) frequency of excitatory neurons in GM6001-treated PTC ($n = 25(5)$) than in nontreated PTC ($n = 27(10)$) and in sham ($n = 36(10)$). **l** Representative Vm traces from nontreated PTC (upper) and GM6001-treated PTC (lower) excitatory neurons on −100pA and 60pA current steps. $n$ = cells(mice) in all. The units for membrane potential (Vm), threshold current, membrane capacitance (Cm) and input resistance ($R_{in}$) are mV, pA, pF, and MΩ, respectively. ****$P < 0.0001$, ***$P < 0.001$, **$P < 0.01$, *$P < 0.05$, one-way ANOVA in (**a**–**d**, **g**–**j**); two-way ANOVA in (**e**, **k**) (GM6001-treated vs. nontreated); Tukey's post-hoc test in all. ####$P < 0.0001$, ###$P < 0.001$, ##$P < 0.01$, #$P < 0.05$, two-way ANOVA, Tukey's post-hoc test in (**k**) (GM6001-treated vs. sham)

Excitatory neurons in GM6001-treated PTC were depolarized (Fig. 7g); however, Cm, $R_{in}$, and threshold current were identical to their nontreated counterparts (Fig. 7h−j). Moreover, the input−output curve showed a lower excitability with small current injections, similar to that of sham, yet still showed somewhat increased excitability with larger current injections than sham (Fig. 7k, l). Note, however, that under these conditions the elevation in glioma-released glutamate remains a driver of heightened excitability in GM6001-treated animals.

Taken together, the peritumoral excitation−inhibition balance is upset as excitatory neurons show a reduced activation threshold, enhanced firing frequency, whereas GABAergic FSNs

show reduced firing frequency, and hence reduced GABA release. Preventing PNN degradation significantly rescued the biophysical characteristics of inhibitory neurons but without affecting excitatory neurons, which were still under the influence of tumor-released glutamate.

**PNNs modulate firing of FSNs.** The above data identified the change in excitability of excitatory neurons as an extrinsic difference attributed to the tumor-released Glu. In contrast, changes in the excitability of FSNs appear intrinsic, and are associated with disintegration of PNNs and ~25% increase in cell

capacitance as the primary differences between PTC and control cortex. If PNN disintegration was sufficient to reduce the firing frequency of FSN, experimental PNN degradation should phenocopy this in control brain slices. As our results attribute peritumoral PNN degradation primarily to glioma-released MMPs (MMP-2, MMP-3, and MMP-9), we used a cocktail of these MMPs to degrade PNNs experimentally. Unfortunately, experimental limitations including incubation duration and low enzymatic activity prevented us to achieve sufficient PNN degradation. We therefore turned to Chondroitinase ABC (ChABC), which is a bacterial enzyme that cleaves polysaccharide chains of N-acetylgalactosamine and glucuronic acid and effectively degrades PNNs (Supplementary Fig. 6a, b).

ChABC-mediated digestion of PNNs in control cortical slices profoundly changed the biophysical properties of FSNs (Fig. 8a) by essentially phenocopying that of FSNs in the PTC; notably ~50% decrease in firing frequency and ~20% increase in Cm (Fig. 8a). No other properties except Vm (Supplementary Fig. 6c) were altered. By contrast, excitatory neurons in ChABC-treated slices exhibited unaltered Cm (Fig. 8b) and other intrinsic properties (Supplementary Fig. 6d), except for a small change in firing frequency (Fig. 8b).

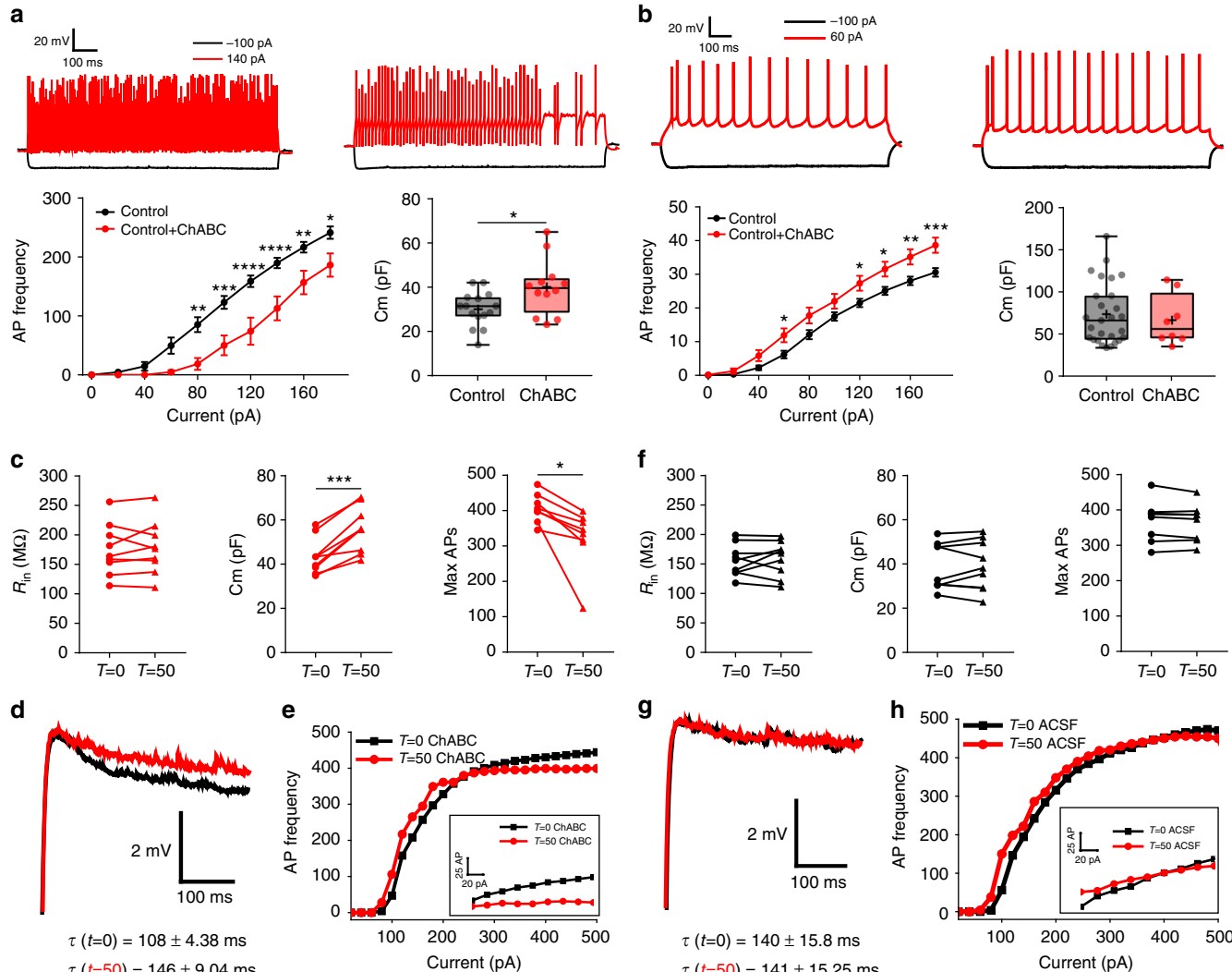

**Fig. 8** PNNs modulate the firing rate of FSNs by reducing the membrane capacitance. **a** Representative Vm traces of FSNs in control (upper left) and ChABC-treated slices (upper right) on −100pA and 140pA current injections. Reduced AP frequency of FSNs in (lower left) ChABC-treated slices ($n = 8$ (6)) than in control ($n = 10$(7)). Higher Cm (lower right) of FSNs in ChABC-treated slices (39.87 ± 3.56, $n = 12$(6)) than in control (30.11 ± 1.99, $n = 15$(8)). **b** Representative Vm traces of excitatory neurons in control (upper left) and ChABC-treated slices (upper right) on −100pA and 60 pA current steps. Higher AP frequency (lower left) of excitatory neurons in ChABC-treated slices ($n = 22$(8)) than in nontreated controls ($n = 36$(10)). Unaltered Cm of excitatory neurons in (lower right) ChABC-treated slices (66.84 ± 10.51, $n = 8$ (6)) than in control (73.93 ± 6.95, $n = 27$(8)). **c** Properties of FSNs during real-time PNN degradation in slices by ChABC ($T = 0$, before ChABC; $T = 50$, 50 min after ChABC superfusion). $R_{in}$ ($T = 0$, 175.05 ± 14.60; $T = 50$, 177.83 ± 14.93; $n = 9$(7)) remained unaltered; however, Cm ($T = 0$, 43.53 ± 2.70; $T = 50$, 55.76 ± 3.43; $n = 9$(6)) and max AP frequency ($T = 0$, 406.62 ± 14.39; $T = 50$, 322.5 ± 30.27, $n = 8$(6)) changed significantly. **d** Representative traces of FSN's Vm decay and **e** AP frequency before (black) and after (red) ChABC treatment. τ = membrane time constant. **f** Electrophysiological properties of FSNs during ACSF superfusion did not change. $R_{in}$ ($T = 0$, 156.07 ± 8.96; $T = 50$, 158.98 ± 9.94; $n = 9$(6)), Cm ($T = 0$, 38.74 ± 3.51; $T = 50$, 39.33 ± 3.75; $n = 9$(6)), and AP frequency ($T = 0$, 368 ± 20.90; $T = 50$, 364.5 ± 18.83; $n = 8$ (6)). **g** Representative traces of FSN's Vm decay and **h** AP frequency before (black) and after (red) 50 min of ACSF treatment. τ = membrane time constant. $n$ = cells(mice) in all. The units for membrane potential (Vm), membrane capacitance (Cm), and input resistance ($R_{in}$) are mV, pF, and MΩ, respectively. ****$P < 0.0001$, ***$P < 0.001$, **$P < 0.01$, *$P < 0.05$, two-way ANOVA, Sidak's post-hoc test in bottom left (**a** and **b**); unpaired $t$ test in bottom right (**a** and **b**); and paired $t$ test in (**c**) and (**f**)

The above data compare neurons in ChABC-treated to untreated brain slices, and therefore reflect differences in properties of FSNs with and without PNNs. To ensure that this is reflective of individual FSN's properties before and after acute loss of PNNs, we also studied biophysical changes of FSNs during acute digestion of their own PNNs. On observance of a randomly patched cell as fast spiking (Supplementary Fig. 1f), we obtained recordings before ($T = 0$) and after superfusion of 1 U/ml ChABC (or only artificial cerebrospinal fluid (ACSF) in control) for 50 min ($T = 50_{ChABC}$) at 32–33 ℃. This approach reliably digested PNNs on the exposed slice surface, yet left sufficient identifiable traces around the recorded FSNs (Supplementary Fig. 7a, b) to infer that the cell initially had an intact PNN.

FSNs did not change Vm either upon ChABC application or during 50 min treatment; however, they showed occasional spontaneous APs and 2–3 mV fluctuations in Vm (Supplementary Fig. 8a, b). FSNs with only ACSF superfusion retained the Vm throughout recordings (Supplementary Fig. 8a, b). The $R_{in}$ of FSNs remained unaltered throughout ChABC (Fig. 8c left) and ACSF superfusions (Fig. 8f left). However, Cm of FSNs increased ~25% on ChABC treatment (Fig. 8c middle) as evident from an increased membrane time constant (Fig. 8d) without changing membrane resistance (Fig. 8c left, Supplementary Fig. 8d). By contrast, Cm of FSNs remained unaffected during ACSF superfusion (Fig. 8f middle, 8g). The firing frequency of individual FSNs after ChABC superfusion (Fig. 8c right) was significantly reduced particularly with stronger stimulation where the maximal firing frequency was reduced by ~20% (Fig. 8e); however, it remained unaltered on ACSF superfusion (Fig. 8f right, 8h). Taken together, these data suggest that an intact PNN reduces membrane capacitance and is essential to allow FSNs to sustain their maximum firing frequency.

ChABC degrades CSPGs in the interstitial matrix and PNNs irrespective of their structural arrangement. To decipher whether CSPGs disintegration influences membrane properties regardless of their structural assembly as PNNs, we measured intrinsic properties of the excitatory neurons that lack PNNs but instead are surrounded by interstitial matrix CSPGs before and after 50 min of ChABC treatment. Excitatory neurons gradually depolarized during ChABC superfusion (Supplementary Fig. 8a, b (right)), a phenomenon we did not study further. However, their $R_{in}$ (Supplementary Fig. 8c, left) and Cm (Supplementary Fig. 8c, third from left) did not show any change.

**PNN regulates FSN firing by modulating specific capacitance.** To investigate whether the experimentally observed Cm alteration is sufficient to explain the firing frequency reduction, we employed a computer simulation to generate a first-order approximation to the Hodgkin−Huxley (HH) differential equation model (Supplementary Fig. 9a, b). We entered empirically recorded values for input resistance and specific membrane capacitance of 1 μF/cm². Maximal firing rate (z-axis) increased nonlinearly with increasing stimulation (y-axis), and decreased progressively with increasing specific membrane capacitance (x-axis). Supplementary Fig. 9b shows a 2D representation of the most relevant capacitance change, suggesting that the observed increase in capacitance estimated ~10% decrease in firing frequency, a value close to the reported ~15–20% decrease (Fig. 8c), and supporting the suggested causal link between the increased specific capacitance after PNN digestion (or between PTC and contralateral) and the decreased maximal firing frequency.

To question whether this change in specific membrane capacitance may also affect the AP discharge pattern at lower firing frequencies, we used Neuronify[36] to model the output of

two interneurons with identical electrical properties. Reducing specific membrane capacitance by 25%, showed ~30 and ~50% increase in the APs numbers, upon stimulation with a random pattern generator at 100 and 40 Hz, respectively (Supplementary Fig. 9c). Hence, these data suggest that not only does the presence of PNN increases maximal sustainable firing frequency, but also the evoked activity is indeed greatly enhanced even at lower frequencies.

**PNN degradation facilitates cortical epileptiform activity.** Finally, to question whether PNN degradation is sufficient to contribute to epileptiform activity, we compared the hyperexcitability latency of excitatory neurons in control and ChABC-pretreated slices on $Mg^{2+}$ removal from the ACSF. This leads to a gradual appearance of short (<200 ms) interictal-like events in both current (Fig. 9a) and voltage (Fig. 9b−d) clamp recordings, followed by delayed appearance of longer lasting (>2 s) ictal-like discharges (Fig. 9b, e). ChABC pretreatment significantly shortened the latency of ictal-like events (Fig. 9e−g) but not interictal-like events (Fig. 9f). Moreover, excitatory neurons gradually depolarized on ChABC superfusion (Supplementary Fig. 8a (bottom), 8b (right)), which may likely contribute to the generation of spontaneous hyperactivity. These results support the notion that the PNN loss is sufficient to facilitate the development of epileptiform hyperexcitability.

## Discussion

This study sought to define changes in peritumoral brain that may contribute to seizure activity in the tumor-associated cortex. Prior studies had already implicated the assiduous release of Glu from the tumor as instrumental, and we indeed find that peritumoral excitatory neurons are significantly depolarized, have a reduced activation threshold and increased discharge frequency but only when proximal to a tumor that expresses the Glu-releasing SXC transporter. Recent studies[3,5] suggested that tumor-released Glu is necessary but not sufficient to cause seizures unless paired with decreased function of GABAergic interneurons. In agreement with prior studies, we observed a 2–5-fold decrease in the PV interneurons density within 0.6 mm of glutamate-releasing tumor suggesting their excitotoxic loss. Interestingly, the surviving PV interneurons show ~50% reduction in firing frequency, a phenomenon that was independent of the tumor-released Glu. By using WFA as a PNN marker that enwrap PV interneurons, we unexpectedly discovered damaged PNNs in conjunction with PV interneuron loss in the PTC. PNN damage was found to be primarily attributed to the proteolytic activity of glioma-released MMPs and consequently caused the cell's membrane capacitance to increase. Experimental digestion of PNNs in normal brain phenocopied the change in capacitance and the observed excitability changes. Further biophysical analysis and modeling results revealed that the PNNs must act as an electrostatic insulator that decreases the cell's effective membrane capacitance and concomitantly increase the cell's maximally attainable firing frequency, which is directly linked to GABA release and maintenance of inhibitory tone.

Hence, in studying glioma-associated seizures, we uncovered a novel and hitherto unrecognized function of PNNs in the normal brain. As illustrated in summary Supplementary Fig. 10, our data suggest that PNN tunes cell's membrane capacitance thereby modulating its firing frequency. We theorize that PNN increases the separation distance between the conducting extracellular and intracellular fluid compartments. The constant negative charge density of PNN's GAGs equilibrates in physiologic electrolyte concentrations to form nonconducting, static charges, viz., a dielectric material. This insulating extension of the lipid bilayer

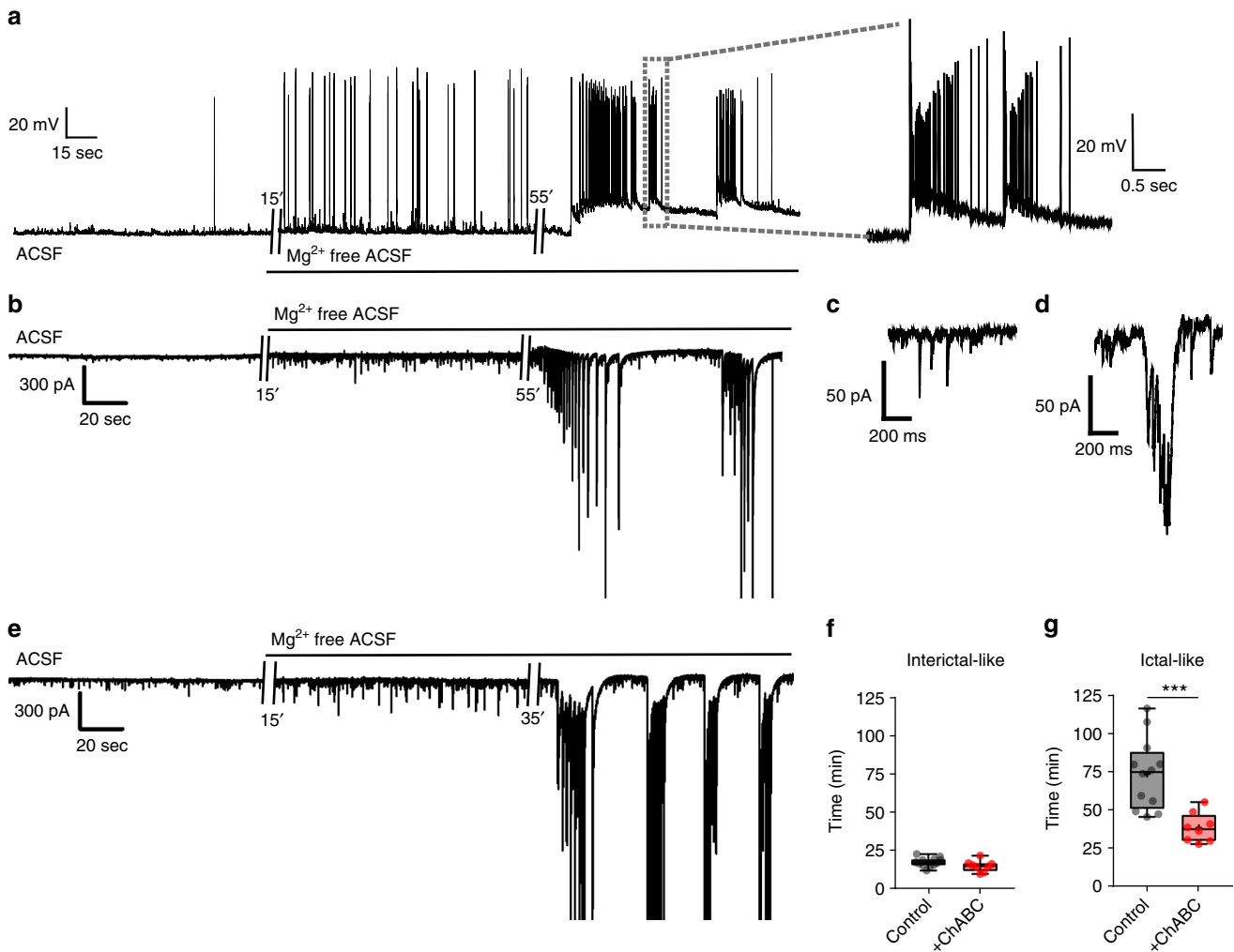

**Fig. 9** PNN degradation facilitates hyperactivity in excitatory neurons. **a** Representative current-clamp ($I = 0$) recording traces from an excitatory neuron showing interictal and ictal-like discharges induced by $Mg^{2+}$ removal from ACSF. Expanded trace shows ictal-like activity. **b−e** Representative voltage-clamp recordings of excitatory neurons from a control (**b**) and ChABC-treated slice (**e**) before and after removing $Mg^{2+}$ from ACSF. The expanded traces in (**c**) and (**d**) correspsond to before and during interictal-like discharges, respectively, in the recording shown in (**b**). **f−g** Latency to $Mg^{2+}$-free-induced (**f**) interictal activity (control, $17.52 \pm 0.81$ min, $n = 12(6)$; ChABC-treated, $15.06 \pm 1.27$ min, $n = 8(5)$; unpaired $t$ test) and (**g**) ictal-like activity (control, $73.67 \pm 6.69$ min, $n = 12(6)$; ChABC-treated, $38.83 \pm 3.40$ min, $n = 8(5)$; ***$P < 0.001$, Welch's $t$ test). $n =$ cells(mice) in (**f**) and (**g**)

should not affect the ionic conductance. Akin to the myelin sheaths surrounding axons, the PNN that extends from the proximal dendrites across the cell body to the axon hillock increases the membrane thickness, which in turn decreases the specific membrane capacitance.

Capacitance of biological membrane is typically considered as standard parallel-plate capacitance, which is inversely proportional to distance of the charge carriers or thickness of the membrane. Therefore, even relatively small changes in thickness have pronounced effect on specific membrane capacitance. The capacitance, in turn, is a limiting factor in charging the axonal membrane and limits the neuronal maximal attainable discharge rate. PV interneurons are among the fastest spiking neurons and their discharge rate appears to be the maximum of physiologically achievable given the constraint of biological membranes and ion channels. The maximal conduction of the $Na^+$ and $K^+$ channels ($G$) and their absolute refractory period are constant for a given cell. Hence the time to charge the membrane $T = 1/G \times C$ can only be further reduced by changing the capacitance ($C$). This in turn lowers the rise time of the action potential ($tr = 2.2\ T$, from 10 to 90% of maximal amplitude). Thus, PNN appears as nature's

trick to allow a cell to fire at supra-physiological levels without any intrinsic changes in neuronal membrane. Indeed, PNNs around hippocampal PV neurons may attribute their relative higher propagation velocity of action potential with rare failure than that of principle neurons under comparable condition[37].

Although PNNs are stable structures and known to lock synapses in place upon closure of the developmental window of plasticity[17,38,39], several CNS pathologies[40], including glioma and epilepsy, present with upregulated ECM and PNN remodeling enzymes including MMPs. Gliomas act as a localized source of seizure provoking Glu and PNN degrading metalloproteinases[28] thereby establishing a gradient of spread, which explains the observed gradient of excitotoxic cell loss and PNN destruction that decreases with distance from the tumor. Reactive astroglial cells in the PTC are also a potential source of PNN degrading agents. However, no spatial correlation between reactive astrocytes and degraded PNNs suggests glioma being the primary source of ECM remolding agents. Note that Glu may also cause an increase in MMP expression and activity level as reported by a recent study[41]. PNNs and CSPGs play crucial roles in protecting neurons from oxidative stress[33] and glutamate excitotoxicity[42]

and combined actions of glioma-released proteases and Glu might explain the selective vulnerability of PV neurons over principle neurons in the PTC.

We suggest that other forms of epilepsies may similarly involve a dysfunction of PNNs. Indeed a number of seizure models show downregulation of the PNN components aggrecan, hyaluronan and proteoglycan link protein 1 (HAPLN1) and hyaluronan synthease-3[43]. Moreover, consistent with our own findings, degrading PNNs using ChABC decreases the threshold for PTZ-induced myoclonic seizures[44]. In addition, depletion of brain hyaluronic acid, which is an important constituent of PNNs, has been shown to generate spontaneous seizures in cats[45].

While we suggest a pivotal role for the MMP activity in glioma-associated epilepsy, heightened MMP activity has also been reported in other models of epilepsy[46] with elevated MMP level being associated with epileptiform activity[47–50]. Moreover, metalloproteinase inhibitors prevented seizures[51] and MMP-9 knockout mice were less susceptible to pentylenetetrazole-induced seizures[52]. Similarly, we show that the MMP inhibitor GM6001 restores normal excitability of FSN by preventing PNN degradation. In addition, GM6001 increased the number of FSNs most likely due to the known neuroprotective effect of the PNNs[33] shielding FSNs from glutamate excitotoxicity[33]. Future studies may therefore examine the antiepileptic potential of MMP inhibitors using a variety of seizure models.

Our study is not the first to implicate the ECM in regulating neuronal excitability in epilepsy; however, it does propose a novel mechanism. Staley and colleagues demonstrated that CSPGs digestion changes the transmembrane $Cl^-$ gradient, which can alter GABAergic neurotransmission[20]. Others have suggested altered lateral mobility of AMPAR[53] or changes in the local diffusion of $Ca^{2+}$ in brain tissue[54] following PNN degradation. The aforementioned mechanisms could alter the excitation−inhibition balance in favor of seizures and may well contribute to a loss of inhibition in tumor-associated cortex. Our experimental data including recording membrane capacitance during real-time digestion of PNNs combined with our in silico modeling studies suggest that, in the case of glioma-associated seizures, the change in specific membrane capacitance upon destruction of the PNNs appears to be the major factor accounting for the enhanced seizure activity. However, it is important to recognize that in addition to changes in the excitability of inhibitory neurons as suggested here and by others[55], PNN degradation may also alter synaptic activity[56] and hence network excitability.

Additional studies on the importance of PNN degradation are warranted; however, our findings have therapeutic implications concerning tumor-associated epilepsy. We propose that blocking matrix degrading enzymes using already approved drugs that pass the blood−brain barrier may be effective in slowing cell invasion and tumor angiogenesis, both processes that require MMP activity[28].

In conclusion, our intention to decipher tumor-associated epilepsy yielded a much bigger finding, namely a novel and hitherto unknown role for PNNs. The dielectric shielding of the membrane that reduces the effective membrane capacitance allows FSNs to achieve supra-physiological discharge frequencies. Other populations of FSNs in brain may be similarly modulated by PNNs, and similarly vulnerable to enzymatic damage of PNNs following injury and disease.

## Methods

**Animals**. All animal procedures were approved and performed in accordance with the ethical guidelines set by Virginia Tech Institutional Animal Care and Use Committee (IACUC). Mice were maintained in groups of five in a specific pathogen-free barrier facility in 12 h light/dark cycles. Male and female C.B.17 scid mice aged 6–9 weeks were used for intracranial tumor implantation. Female

athymic nude mice aged 6–8 weeks were used for flank injections, maintenance and propagation of the GBM xenografts. Astrocyte-specific β1-integrin knockout FVB-N mice were generated as described previously[31].

**Patient-derived xenograft tumors**. Patient-derived primary glioma tissue was maintained by serial passage in nude mice flanks as described previously[4]. Briefly, tumors were harvested after 14–18 days postinjection and glioma cells were mechanically and enzymatically dissociated. Cells were passed through a 40 μm filter and maintained as "gliospheres" in Dulbecco's modified Eagle medium/nutrient mixture F-12 (DMEM/F-12), supplemented with 10 mg/ml fibroblast growth factor (FGF), 10 mg/ml epidermal growth factor (EGF), 260 mM L-glutamine, 2% B-27 Supplement without vitamin A (Invitrogen), 250 μM/ml amphotericin, and 50 mg/ml gentamycin (Fisher), and incubated in 10% $CO_2$ at 37 °C. Medium was changed daily for 2 days, then weekly. Gliospheres were maintained in vitro for 5–7 days before intracranial injection. For intracranial injections, cells were dissociated with Accutase (Sigma-Aldrich), counted, and then diluted in sterile phosphate-buffered saline (PBS) to get desired cells/unit volume.

**Intracranial glioma injections**. Human glioblastoma GBM14 and GBM22 tumors, previously established from human biopsies, were implanted into 6–9 weeks old immunodeficient C.B.17 scid mice as previously described[4]. Briefly, mice were anesthetized with 2–5% isoflurane and fixed to a stereotaxic apparatus (Leica Angleone stereotaxic model 39464710) followed by a midline scalp incision and a 0.5 mm burr hole 1.0–2.0 mm lateral and 0.5–1.0 mm anterior to bregma. Patient-derived xenograft tumor cells ($2.0 \times 10^5$ cells in 2 μl of PBS, GBM22 and GBM14) were injected at a depth of 2.0–2.5 mm. Control mice were injected with sterile PBS. Body weight of animals was measured on alternate days and experiments were conducted between 12 and 20 days post-glioma implantation. To specifically target glioma cells to grow near dorsal hippocampus, we injected glioma intracranially at 1.25 mm lateral, 1.46 mm posterior and 1.4 mm ventral from bregma. A 10 μl syringe (World Precision Instruments #SGE010RNS) was used to infuse glioma cells at 11 nl/s rate.

**GM6001 injections, cryofixation and in situ zymography**. GM6001 or illomastat (ApexBio Cat# A4050), a broad-spectrum MMP inhibitor (Ki: 0.4 nM for MMP-1, 27 nM for MMP-3, 0.5 nM for MMP-2, 0.1 nM for MMP-8 and 0.2 nM for MMP-9), was administered intraperitoneally as a suspension of 100 mg/kg/day in 4% carboxymethylcellulose (CMC) in saline to a total volume of 200 μl[57]. Mice in the control group received 4% CMC in saline. Injection of GM6001 or CMC (control) was initiated 2 days after glioma implantation to let mice recover and allow glioma to settle in brain, followed by additional injections every 24 h for total 13 days. GM6001/CMC-treated mice were sacrificed to harvest brains for in situ zymography on 14 days after glioma implantation. Mice were deeply anesthetized with intraperitoneal injection with a cocktail of ketamine (100 mg/kg of body weight) and xylazine (10 mg/kg of body weight) followed by transcardial perfusion with ice-cold PBS for 2 min and a quick dissection of brain in ice-cold N-methyl-D-glucamine (NMDG)-based cutting solution. To achieve rapid freezing of tissue without forming ice crystals, we trimmed out one-fourth brain from caudal side to decrease the tissue size. Brain tissues were immersed in tissue freezing medium (TFM5, Electron microscopy Sciences) in a mold (4566 Tissue tek Cryomold) followed by placing the mold on 2-propanol-filled beaker surrounded by dry ice. The total tissue freezing time from brain mounting in TFM to its solidification did not exceed one and half minute, which ensured the preservation of enzymatic activity in the tissue. Sections were stored at −80 °C and moved to −20 °C for overnight before cryo-sectioning. Coronal cryo-sections (15 μm) were cut using cryostat (Leica CM 1850 UV) and mounted on glass slides (Superfrost plus microslides, Cat# 48311-703). For electrophysiology experiments, GM6001-treated mice were used (one mouse per day) from 14th day post-glioma implantation and the remaining mice were routinely injected with GM6001 until used.

In situ gelatinolytic activity of MMP-2/9 was performed on above-mentioned frozen sections using a commercial kit (EnzChek Gelatinase Assay kit, Molecular Probes, Cat# E-12055). In brief, sections were thawed on ice for 30 min and followed by 3–5 washes with PBS to remove TFM. Sections were incubated with DQ gelatin fluorescein conjugate (20 μg/ml), at 37 °C for 1 h in a dark and humid chamber to optimize gelatinolytic activity. Cleavage of gelatin by active tissue MMPs exposes the fluorescein molecules (495/515 nm). Substrate concentration was optimized for the assay and 20 μg/ml was used for all the experiments. 1,10-phenanthroline (1 mM in dimethyl sulfoxide (DMSO), Sigma-Aldrich) was used as a nonspecific inhibitor of MMP (Supplementary Fig. 2c). To remove excess fluorogenic substrate, sections were washed 3–5 times and fixed in 4% paraformaldehyde (PFA) for 5 min. Further, sections were incubated with primary antibody specific for astrocytes (Chicken GFAP, Cat# ab4674, Abcam, 1:1000) and neurons (Rabbit NeuN, Cat# ABN78, Millipore, 1:500; Mouse PV, Cat# PV 235, Swant, 1:1000), and biotinylated WFA (Vector laboratories, Cat# B-1355, 1:500) for 1 h at room temperature. Appropriate fluorophore-conjugated secondary antibodies and Alexa Fluor® 555-conjugated streptavidin (Vector laboratories, Cat# S32355, 1:500) were used to detect the primary antibodies and WFA, respectively. Sections were examined and images were taken using Nikon A1 confocal microscope within 48 h of staining procedure. Images were acquired at various

magnifications and data were quantified using Nikon Elements analysis program associated with Nikon A1 confocal microscope.

**Acute slice electrophysiology.** After cervical dislocation, mice were quickly decapitated and brains were dissected out and kept in an ice-cold ACSF (135 mM NMDG, 1.5 mM KCl, 1.5 mM $KH_2PO_4$, 23 mM choline bicarbonate, 25 mM D-glucose, 0.5 mM $CaCl_2$, 3.5 mM $MgSO_4$; pH 7.35, 310 ± 5 mOsm) (Sigma-Aldrich) saturated with carbogen (95% $O_2$ + 5% $CO_2$). Coronal slices (300 μm) were prepared using Leica VT 1000P tissue slicer and slices were allowed to recover for 1 h in ACSF (125 mM NaCl, 3 mM KCl, 1.25 mM $NaH_2PO_4$, 25 mM $NaHCO_3$, 2 mM $CaCl_2$, 1.3 mM $MgSO_4$, 25 mM glucose, pH 7.35, 310 ± 5 mOsm) at 32 °C. Afterwards slices were kept at room temperature until used for recordings. Individual slices were placed in a recording chamber continuously superfused with ACSF at a flow rate of 2 ml/min. Glioma cells grow rapidly and expand to primary motor and somatosensory cortical areas after 12–14 days postinjection. Tumor mass in these cortical areas was visually identified by their unique appearance as described previously[6]. The pyramidal cells and FSNs in cortical layers 3–5 were identified under an upright microscope (Leica DMLFSA) with ×40 water immersion lens and infrared illumination. Whole-cell voltage-clamp and current-clamp recordings were achieved using an Axopatch 200B amplifier (Molecular Devices). Patch pipettes of 3–5 MΩ open-tip resistance were created from standard borosilicate capillaries (WPI, 4IN THINWALL GI 1.5OD/1.12ID) using Narishige PP-83 and HEKA PIP 6 vertical pipette pullers. Patch pipettes were filled with an intracellular solution of 134 mM potassium gluconate, 1 mM KCl, 10 mM 4-(2-hydroxyethyl)-1-piperazineethanesulfonic acid (HEPES), 2 mM adenosine 5′-triphosphate magnesium salt (Mg-ATP), 0.2 mM guanosine 5′-triphosphate sodium salt (Na-GTP) and 0.5 mM ethylene glycol tetraacetic acid (EGTA) (pH 7.4, 290–295 mOsm). Unless otherwise stated, we added 20 μl Lucifer yellow (Sigma Cat# L0259, 20 mg/ml stock solution in deionized water) in intracellular buffer just before the recording for post-experiment identification of the cells. Patch pipettes were visually guided using MM-225 micromanipulator (Sutter Instrument, Navato, CA). A potential damage to the PNNs was minimized during the patching procedure by applying minimum positive pressure while approaching cells. Whole-cell recordings were made once >5–10 GΩ seal was achieved. For voltage-clamp recordings, the membrane potential was clamped at −70 mV. The membrane capacitance (Cm) and series resistance were not compensated unless otherwise stated. For few experiments, we superfused ACSF containing 50 μM D-2-amino-5-phosphonovalerate (D-AP-5) and 20 μM 6-cyano-7-nitroquinoxaline-2,3-dione (CNQX) to block glutamatergic neurotransmission and confirmed the blockage by observing disappearance of spontaneous EPSCs. Data were acquired using Clampex 10.4 software and Axon Digidata 1550A interface (Molecular Devices), filtered at 5 kHz, digitized at 10–20 kHz and analyzed using Clampfit 10.6 software (Molecular Devices). Unless otherwise stated, during all the recordings carbogen-bubbled ACSF was continuously superfused (2 ml/min) and all recordings were made at 32–33 °C using an inline feedback heating system (Cat# TC 324B, Warner Instruments).

Measurement of intrinsic properties: The resting membrane potential (Vm) was measured by setting $I = 0$ mode immediately after achieving whole cell. Action potential (AP) threshold current was calculated by injecting 2−200 pA current pulses (10 ms duration) with 2 pA increment in each step. Minimum current required to generate first AP where onwards each subsequent higher current leads to AP generation was noted as threshold current. Most of the cells fired APs within this current range; however, cells that did not fire within this range were lumped in single group of 200 pA firing threshold. To calculate input resistance ($R_{in}$), we injected 15 hyperpolarization current steps (−100 pA each for 1000 ms) and recorded the steady-state membrane voltage deflection (ΔV). The $R_{in}$ was measured as a ratio (ΔV/I) of steady-state change in the membrane voltage (ΔV) and the corresponding injected current (I). Membrane capacitance (Cm) was also calculated from the same 15 voltage responses of hyperpolarization current steps. All 15 voltage traces were averaged and the discharging phase of voltage was fit with a simple exponential decay and the slowest time constant was used for calculating the Cm[58]. The excitability of neurons was assessed using the input−output curve obtained by applying increasing step currents of different magnitudes (−100 to 180 pA, 15 steps with 20 pA increment each step, step duration 1000 ms) and counting number of APs using Clampfit 10.6 program. We observed that the amplitudes of APs shortened as the firing frequency increased; therefore, we set a minimum 15 mV deflection from the steady-state response as a qualifying criterion for a spike to be identified as an AP. To check the saturating/maximum firing frequency (AP frequency) supported by FSNs in our experimental conditions, we kept increasing the magnitude of injected current until the firing frequency did not increase any further with the higher currents. The observed saturating/maximum firing frequency was found to be achieved by injecting 400 −500 pA; therefore, we capped the highest injected current to 500 pA.

**Identification of FSNs with PNNs in peritumoral cortex.** As evident by our immunohistochemical staining results, disintegration of PNNs was reliably found within ~0.5 mm of the tumor border. We randomly patched neurons within this range and later confirmed the disintegrated status of PNN by WFA staining and measuring the distance of the patched cell from the growing border of glioma. (Supplementary Fig. 3a, b). We added Lucifer yellow dye in the internal patch

solution for postrecording identification to determine the structural integrity of PNNs around the recorded cell and the distance from the tumor border. On completion of recordings, patch pipette was carefully retracted to minimize the membrane damage and slices were fixed in 4% PFA overnight and later stained for WFA and DAPI to check the presence of PNNs around the patched neurons (Supplementary Fig. 3). The slices were incubated with biotinylated WFA (1:200) for 1.5 h followed by three rinses with PBS, and subsequently incubated for 1 h with Alexa Fluor® 555-conjugated streptavidin (Vector laboratories, Cat# S32355, 1:200). Then, the slices were rinsed thrice with PBS, stained with DAPI for 3 min (Cat# D1306, Life Technologies, 1:1000 diluted from 2 mg/ml stock) and mounted on the glass slides using mounting medium (SlowFade Gold antifade reagent, Cat# S36936, Invitrogen) and cover glasses (20 × 50-1, Cat# 12-548-5E, Fisher). The distance of patched cell from the tumor border was also measured in the images and only those cells falling within 500 μm of tumor border were used for analysis. Glioma mass was identified as densely packed amorphous mass of DAPI-positive cells (Supplementary Fig. 3).

**Acute slice PNN degradation and identification of $PNN^+$ cells.** Chondroitinase ABC (ChABC) from *Proteus vulgaris* (Cat# C3667, Sigma-Aldrich) was reconstituted in a 0.01% bovine serum albumin aqueous solution according to the manufacturer's instruction to make 1 U/40 μl stock solution. Aliquots of 2 U were prepared and stored at −20 °C until used. We performed ECM/PNN degradation by two methods depending on the experimental requirement.

For one set of experiments, we pretreated recovered slices with ChABC and thereafter performed recordings on the ChABC-treated slices. In brief, on completion of post-slicing recovery, 2–3 cortical half slices were incubated in 3 ml of 0.5 U ChABC per ml of ACSF in an incubation chamber continuously supplied with carbogen at 33 °C for 45 min. Then, the slices were rinsed with and incubated in ACSF until used for electrophysiological recordings. These parameters of PNNs digestion by ChABC (enzyme concentration—0.5 U/ml, incubation time—45 min, incubation temperature—33 °C) reliably degraded PNNs but spared traces around the FSNs thereby allowing the identification of PNN-positive cells (Supplementary Fig. 6a, b). Similarly, previously separated contralateral halves of the ChABC-treated slices were incubated in 3 ml of ACSF without ChABC in the incubation chamber. Then, both the ChABC-treated and nontreated slices were kept in ACSF together until used for the recordings.

For the second set of experiments, we performed real-time digestion of PNNs in acute slices simultaneously with the electrophysiological recordings. We randomly patched FSNs and recorded their intrinsic properties and subsequently monitored their baseline membrane voltage (I = 0) for 5–10 min. The cells with resting membrane potential unstable and greater than −60 mV were discontinued. After baseline membrane potential recording, we superfused 1 U/ml ChABC solution for 50 min while strictly maintaining the bath temperature at 32–33 °C as it appeared to be critical for PNN degradation. This protocol also reliably degraded PNNs but left identifiable traces around the FSNs to confirm the presence of PNNs around patched cells prior to the digestion (Supplementary Fig. 7). On completion of recordings, slices were fixed in 4% PFA overnight followed by WFA staining as described above. The FSNs expressing PNNs in the ChABC-treated slices were identified by a low WFA intensity and disintegrated PNN around the cell body of LY-filled cell (Supplementary Fig. 7a, ChABC) compared to their non-ChABC-treated counterpart exhibiting intense WFA staining around the cell body and proximal dendrites of LY-filled cell (Supplementary Fig. 7a, ACSF). The recorded excitatory neurons appeared as the LY-filled cells with intact PNNs and high WFA intensity background in nontreated slices (Supplementary Fig. 7b, ACSF) and as the LY-filled cells with nearby low WFA intensity and with only identifiable traces of PNNs in ChABC-treated slice (Supplementary Fig. 7b, ChABC).

**Immunohistochemistry.** Animals of different experimental groups were injected with a mixture of ketamine and xylazine (100 mg/kg and 10 mg/kg, respectively) and subsequently trancardialy perfused with PBS followed by 4% PFA. The brains were dissected out and stored overnight in 4% PFA at 4 °C. Next, the brains were transferred to PBS for 48 h and 50-μm-thick floating sections were cut using vibratome (5100MZ, Campden instruments). The sections were either used for IHC immediately or stored at −20 °C in a custom-made cryoprservative medium (10% (v/v) 0.2 mM phosphate buffer, 30% (v/v) glycerol, 30% (v/v) ethylenglycol in deionized water, pH 7.2–7.4) for later uses. To minimize procedure-related variability, we preformed staining in a large batch of duplicate sections from 5 to 7 mice of each treatment group. The sections were retrieved from −20 °C storage, rinsed with PBS, and permeabilized and blocked by incubating in blocking buffer (0.5% Triton X-100 and 10% goat serum in PBS) for 2 h at RT. Then the sections were incubated overnight at 4 °C with appropriate primary antibodies and biotinylated WFA (Cat# B-1355, Vector Laboratories) in diluted blocking buffer (1:3 of blocking buffer and PBS). On the next day, the sections were incubated with appropriate secondary antibodies and Alexa Fluor® 555-conjugated streptavidin (Cat# S32355, ThermoFisher Scientific, 1:500) in diluted blocking buffer for 2 h at RT in dark. Further, the sections were rinsed with PBS and occasionally stained with DAPI (10 μg/ml in PBS for 3 min) followed by two rinses with PBS. The sections were mounted on the glass slides (Fisherfinest 25 × 25 × 1, Cat# 12-544-2) covered with cover glass and the edges of slides were sealed with nail polish. The antibodies used were rabbit NeuN (Cat# ABN78, Millipore, 1:500), mouse PV

(Cat# PV 235, Swant, 1:1000), and chicken GFAP (Cat# ab4674, Abcam, 1:1000). Images were acquired at different magnifications using Nikon A1 confocal microscope and quantification was performed by inbuilt NIS-Elements AR analysis program. High-magnification images in all three panels of Figs. 2a and 4e are snapshots of 3D volume images of 30–40-µm-thick z-stacks. Figure 5f and Supplementary Figs. 1b, 3b (panel images), 3c, and 2a (top right) are snapshots of 3D volume image of ~10-µm-thick z-stacks.

**Cell counting.** NeuN$^+$, PV$^+$ and PNN$^+$ cells were counted from 100 µm × 100 µm square region in ×10 magnification single plane images. In brief, a grid of 100 µm × 100 µm boxes was drawn and the boxes containing both glioma mass and brain parenchyma were defined as the tumor border. The number of cells in each subsequent square away from the tumor border was counted and arranged with respect to their distance from the tumor border and their cortical layer location. Each cortical layer has characteristic density of neurons; therefore, spatial localization of glioma in a specific cortical layer(s) may affect the cell density differently. To overcome these variations we analyzed cortical sections in which glioma was present in all the layers and counted cells in 100 µm × 100 µm boxes in each layer in all the comparative test groups such as peritumoral, contralateral, and sham. We restricted the analytical comparisons to layers 3–5 due to highest density of PV/WFA-expressing neurons. Average number of cells per 0.01 mm$^2$ area (corresponding to 100 µm × 100 µm box) for each spatial bin in a brain slice was calculated by averaging the numbers of cells in 3–5 boxes. Similarly, average numbers of cells per 0.01 mm$^2$ area for each spatial bin were calculated from 10 to 15 brain slices from minimum five mice (minimum two brain slices from each mouse) and averaged to obtain mean ± SEM and the numbers of analyzed brain slices were considered as $n$ for statistical analysis. To compare the relative cell density of excitatory and inhibitory neurons in the PTC and contralateral cortex, we normalized them with their respective cell densities in the cortex of sham-treated mice.

**Quantification of IHC and ISZ images.** To evaluate the glioma-induced PNN/interstitial matrix degradation, we quantified the IHC data of WFA intensity in *stratum pyramidale* and *stratum radiatum* of hippocampal CA2 region and PTC. In brief, we averaged the mean fluorescence intensities of randomly drawn 4–6 regions of interest (ROIs) of 15 µm × 15 µm area in each slice. In the PTC areas, we placed these ROIs onto individual PNNs to overcome the potential discrepancy due to different density of PNNs and neurons in similar dimension areas. Subsequently, the mean intensities from 8 to 12 brain slices from minimum five mice were averaged to obtain mean ± SEM and the numbers of analyzed brain slices were considered as $n$ for statistical analysis. To quantify gelatinase activity (fluorescein intensity) and WFA intensity in the ISZ experiments, we drew concentric circles in the PTC at every 200 µm starting from the glioma border to spatially bin the distance-dependent effect of glioma. Next, we drew random 3–5 ROIs of 20 µm × 20 µm dimension in each 200 µm bin area and averaged it to obtain mean intensity which was considered as $n$. Mean fluorescence intensities of each area from 2 to 3 slices per animal (5–7 mice in each treatment group) were used to calculate mean ± SEM for graphical representation and statistical analysis.

**Line profile intensity and PNN integrity analysis.** To quantify the structural integrity of PNNs, we acquired high-magnification (40 × 5) single plane images of individual PNNs in different experimental groups. A polyline was drawn along the entire periphery of cell (stained with NeuN/PV/gelatinase) with PNNs (Fig. 2c). WFA fluorescence intensity of this line shows high intensity peaks and low intensity drops on regular intervals representing areas covered by PNN-CSPGs and holes in the PNNs, respectively. We set a threshold of ~50% of highest fluorescence intensity (red two-headed arrow in Fig. 2d) and counted the number of peaks above the threshold for 10–20 PNNs in each experimental group using Clampfit 10.6 program. A line profile intensity graph showing almost flat line with only few low intensity peaks indicates disintegrated PNN by glioma as exemplified in Fig. 2d. This scheme was adopted to generate data in Figs. 2e and 5h. For representing different stages of PNN and cell disintegration in the PTC in Supplementary Fig. 1d, we drew a straight line across the cell in different areas of the PTC, and the fluorescence intensities of WFA, PV, NeuN and DAPI along this line were plotted. Different patterns of PV and NeuN intensity peaks flanked by WFA peaks indicate whether a cell is present inside the PNN and the status of PNN.

**HH modeling of PNN.** We developed a first-order HH differential equation model using Matlab. This model was first described in the seminal work by Hodgkin and Huxley, and modified by Abbot and Kepler[59,60]. We used the same parameter values as Abbot and Kepler[59]; resting membrane potential, $V_R$(−65 mV); sodium reversal potential, $E_{Na}$ (50 mV); potassium reversal potential, $E_k$(−77 mV); leak potential, $E_l$ (−54.4 mV); specific membrane capacitance, Cm (1 µF/cm$^2$); maximum potassium conductance, $G_k$ (36e$^{-3}$/ohm/cm$^2$); maximum sodium conductance, $G_{Na}$ (120e$^{-3}$/ohm/cm$^2$); leak conductance, $G_l$ (0.3e$^{-3}$/ohm/cm$^2$); external applied current density, $J_{ext}$ (0.2e$^{-4}$ A/cm$^2$); and temperature (18.5 °C). This similar methodology, both model interactions and parameter values, has been used in recent studies[61–63]. A step size of 1 µs was used for the numerical method.

Briefly, Hodgkin and Huxley modeled the single-cell action potential using nonlinear voltage-dependent channel resistors in series with the respective ionic

Nernst potential to charge a parallel membrane capacitor. This circuit model results in the following differential equation:

$$dV_m(t)/dt = 1/Cm(J_{ext} - J_{Na} - J_k - J_{leak}),$$

$$dV_m(t)/dt = 1/Cm(J_{ext} - g_{Na}(V_m - E_{Na}) - g_k(V_m - E_k) - J_{leak}).$$

The sodium and potassium current through their respective variable resistor are governed by three gating variables: $n$ for potassium, $m$ for sodium deactivation, and $h$ for sodium inactivation.

$$g_k(t) = G_k m^4(V_m) \text{ and}$$

$$g_{Na}(t) = G_{Na} n^3(V_m) h(V_m),$$

where $g_k$ and $g_{Na}$ denote the time-varying potassium and sodium conductance, respectively.

The channel gating variables approach their steady-state values according to fitted equations using channel population on ($\alpha$) and off ($\beta$) probabilities.

$$dn(t)/dt = \alpha_n(1 - n) - \beta_n(n),$$

$$dm(t)/dt = \alpha_m(1 - m) - \beta_m(m),$$

$$dh(t)/dt = \alpha_h(1 - h) - \beta_h(h),$$

where,

$$\alpha_n = \phi(0.10 - 0.01\,dV)/(\exp(1 - 0.1\,dV) - 1),$$

$$\beta_n = \phi(0.125)(\exp(-dV/80)),$$

$$\alpha_m = \phi(2.5 - 0.1\,dV)/(\exp(2.5 - 0.1\,dV) - 1),$$

$$\beta_m = \phi(4)\exp(-dV/18),$$

$$\alpha_h = \phi(0.07)\exp(-dV/20),$$

$$\beta_h = \phi(1)/(\exp(3.0 - 0.1\,dV) + 1),$$

$$\phi = 3^{((T-6.3)/10)}; dV = (v - Vr) \times 1000.$$

The initial condition for all simulations was: $(Vr, n(0), m(0), h(0)) = (-65\,mV, 0.3177, 0.0529, 0.5961)$.

Each simulation sampled from a (capacitance, external current) space. The capacitance ranged from 0.8 to 1.5 µF/cm$^2$ with a step size of 0.01 µF/cm$^2$; the external current ranged from 0.2$^{-4}$ to 1.4$^{-2}$. The firing frequency as a function of capacitance and external current was calculated by using Matlab's *findpeaks* function on the voltage time series output for each simulation. Firing frequency was calculated as the number of peaks >20 mV in the 1000 ms simulation.

**Statistical analysis.** Data are represented as box and whisker plots unless otherwise stated in the specific figures. The central lines and plus signs in box and whisker plot represent the medians and means, respectively; the two ends of the rectangles represent first and third quartiles. The upper and lower whiskers extend to the highest and lowest values in the data set, respectively. Individual data points are represented by dots. Statistical details of experiments are mentioned in the figure legends and all the details of statistical analysis including test statistics, $P$ values, post-hoc comparisons, and 95% confidence intervals of mean differences are summarized in Supplementary Table 1. Data are sufficiently normal distributed and variance within groups is sufficiently similar to be used for parametric tests. Experimental designs with two treatment groups were analyzed by two-tailed unpaired or paired $t$ test. Welch's correction was applied where variances of both the groups were statistically different. Experimental designs with more than two groups were analyzed using one-way ANOVA or two-way ANOVA followed by Tukey's or Sidak's post-hoc multiple comparison tests. Statistically significant difference between groups were notified in graphs using asterisk(s) (*$P < 0.05$, **$P < 0.01$, ***$P < 0.001$, ****$P < 0.0001$) and occasionally number sign (#$P < 0.05$, ##$P < 0.01$, ###$P < 0.001$, ####$P < 0.0001$). Data analysis was performed using GraphPad Prism 7.0, Microsoft Excel, and Origin 2016 (OriginLab).

**Code availability**. Matlab codes will be provided on reasonable request to the corresponding author.

## Data availability

All the relevant data is present in the manuscript and the additional details will be provided on reasonable request to the corresponding author.

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

## Acknowledgements

We acknowledge Emily G. Thompson for valuable discussions on in situ zymography, Dr. Ian F. Kimbrough for valuable discussions on image analysis, Paul Youmans for technical assistance and Dr. Kristin F. Phillips for editorial advice. Grant support: NIH-RO1-NS036692, NIH-RO1-NS082851, NIH-RO1-NS052634.

## Author contributions

B.P.T.—designed, performed and analyzed electrophysiology and IHC experiments, wrote the manuscript. L.C.—designed, performed and analyzed ISZ experiments, glioma maintenance and implantation, editorial advice. S.L.C.—experimental design and supervision, editorial advice. D.C.P.—Glioma implantation, statistical analysis, manuscript preparation, editorial advice. A.E.G.—computer simulation, editorial advice. H.S.—conceived idea, experimental design, analysis and interpretation, wrote the manuscript, computer simulations, project supervision.

## Additional information

**Competing interests:** The authors declare no competing interests.

