## [Peer Review File · Nature Communications]

Reviewers' comments:

Reviewer #1 (Remarks to the Author):

This study by Tewari et al. builds on previous reports from their and other labs that brain tumours are associated with epilepsy and that this not only involves excess Glu release by tumor-related cells but possibly also loss of GABAergic interneurons (particularly fast-spiking neurons (FSNs) and their associated perineuronal nets (PNNs)). Here they address directly the FSN and PNN loss in mouse brain tumour models. They show that loss of PV interneurons and PNNs is particularly strong when epileptogenic gliomas are used in mice, and that the PNN loss is correlated with gelatinase activity and reduced by inhibitors of gelatinase. They then show that FSNs exhibit greatly reduced firing frequencies in the vicinity of gliomas (both epileptogenic and non-epileptogenic), and that disruption of PNNs in slices leads to reduced firing of PNNs and enhanced burst firing by excitatory neurons. The authors use modelling to suggest a mechanism in which PNNs have myelin-like properties, enhancing the excitability of FSNs.

This study provides interesting observations relating brain tumours, FSN dysfunction and PNN loss, and suggesting possible causal relationships between these processes and epileptogenesis induced by brain tumours. They further provide potentially important insights as to the role of PNNs in FSN and network function. However, the evidence relating causally PNN loss, FSN dysfunction and epileptogenesis induced by tumors is not compelling. Notably, there seems to be an important discrepancy between how non-epileptogenic gliomas have only a modest effect on PV and PNN loss compared to epileptogenic ones (Fig. 3), and the fact that both types of gliomas have strong and undistinguishable effects on FSN firing rates in situ. Furthermore, the study ignores a vast literature that has related degradation of PNNs to enhanced plasticity and brain repair, without detrimental effects to network activity in situ (including in settings such as stroke models, where seizures can be an issue).

Overall, this study reports on a set of interesting observations concerning FSNs and PNNs, but their significance for tumor-induced epilepsy and for PNN function is not entirely clear.

Specific points:

1) In the opinion of this Reviewer, one main issue concerns a comparison of the findings in Fig3 (epileptogenic gliomas are much more effective at inducing PV and PNN loss than non-epileptogenic ones) and in Fig. 7 (undistinguishable effects of epileptogenic and non-epileptogenic gliomas on the (reduced) firing properties of FSNs in situ). These combined data suggest that reduced FSN firing is poorly correlated to PNN loss in situ, and not correlated to how gliomas induce epilepsy. This appears to contradict some of the main conclusions of the study.

2) The key data relating gelatinase activity inhibition to PNN protection in situ are not as compelling as one would like them to be. Gelatinase inhibitors only have a modest effect on preventing PNN disassembly (Fig. 6). In these experiments, gelatinase activity was not completely inhibited, which might explain the modest protective effect on PNNs. However, since this is such a central point of the study it would be important to better relate the two phenomena: can gelatinase be inhibited more completely and does that then result in complete PNN protection? And what about epileptic activity induced by the glioma: is it also dependent on gelatinase activity?

3) The authors equate PV signal loss to loss of PV neurons. In other settings, particularly schizophrenia, this has been shown not to be accurate: there can be a great reduction/loss of PV signals without loss of PV neurons. The authors find that PNN and PV neurons are affected at a greater distance from the tumor than are NeuN+ total neuron numbers. Possibly, the distant effects might reflect loss of PNN and PV protein without neuronal loss, which might be interesting.

4) The authors ignore a large number of studies that have related local PNN degradation in the

adult brain to highly beneficial effects on brain plasticity and repair (and on sensitive periods for plasticity). How might those findings relate to their results here?

5) The discussion is largely devoted to speculations regarding how organised ECM in PNN might affect excitability, and how PNN loss might relate to epilepsy. What is missing, however, is a critical treatment of the findings in the paper, and of the extent to which the key conclusions are based on compelling evidence.

Reviewer #2 (Remarks to the Author):

The current manuscript by Tewari et al. aims to describe novel mechanisms associated with the pathophysiology of tumor-associated epileptogenesis. Here, the authors provide evidences (but see major comments) specific to epileptogenic glioma GBM22 that support the two components of loss of GABAergic inhibition required for peritumoral epilepsy: 1) loss of PV interneurons due to excitotoxic levels of glu released by glioma, and 2) electrophysiological dysfunction of PV interneurons and loss of perineuronal nets partly due to glioma-secreted MMPs. Interestingly, the authors show that PNN degradation alone is enough to significantly reduce the firing rate of PV interneurons, enough to disrupt GABAergic tone. While some of the findings are interesting, there are major issues with this study. Most importantly, Glioma model data and ChABC model are not logically well connected (see major comment 2) giving an impression that two independent studies (Glioma and ChABC) are presented together with weak logical connections at least in this current form. In addition, experimental data/analysis is very weak to support Glu-independent/MMP-dependent impact to PNNs integrity (see major comment 1). Given that MMP is already known to mediate glioma induced seizure, it is also essential to more thoroughly link MMP and PNNs properties (see major comment 5).

Major Comments:

#1: One of the major points of this study is the MMP dependent (Glu independent) impact of Glioma to PNNs (independent to cell loss). However, there is no quantitative data supporting this point. While authors quantified WFA intensity, this readout does not distinguish PV cell loss vs PNN loss without cell death. While authors show qualitative representative images of PNNs integrity, there is no quantification. Without through quantification of PNNs integrity and analysis taking into account for the PV cell loss (doublestaining of WFA and PV), the major point of this manuscript is not supported. Specifically, there seems to be no detailed methodological explanation for the process of line intensity profiling of cells in the experiments for Figure 2b-i and Figure 6 d-h. Moreover, there are no statistical comparisons involved that would show significance to the authors' claims. Specifically, in terms of Figure 2, "few viable neurons in far peritumoral zone..." – how many (#/area), and what is the exact distance? "Majority of neurons surrounded by invading glioma cells..." – how much (#/area), how many glioma cells = "surrounded"? Similarly, for Figure 6, the number of neurons observed is also not indicated. For consistency, the representative images in 6d-g should also include DAPI and gelatinase.

#2: Another major claim of this study is that FSI firing frequency change in glioma model is due to changes in capacitance because of PNNs degradation. However, this is not supported by the results. Firing frequency in GBM22 and GBM14 groups changed equally but glioma GBM14 and glioma GBM22 affected capacitance differently (fig 7i vs supplemental fig 4f). Capacitance was not impacted by GBM14 suggesting that capacitance difference in GBM22 is likely the consequence of chronic exposure to glu from glioma. Given this dissociation between glioma model and ChABC model, there is a logical flaw in presenting glioma related data and ChABC data together in the same manuscript. Authors also failed to cite published work showed ChABC decrease FSI excitability (Balmer eNeuro 2016).

#3: It is not clear whether the firing frequency (fig 7e, 7i) are measured with or without synaptic

blockers. As the amount of both excitatory and inhibitory cells change near glioma, it is obvious to expect that the number of synaptic contacts and synaptic drive changed. To support authors claim that changes shown in 7e (for pyramidal neurons) are extrinsic and 7f (for FS interneurons) are intrinsic, recording with synaptic blockers are necessary to support the authors' interpretation.

#4: It would be very important to thoroughly assess the "causal" contribution of MMP to the properties of PNNs. While WFA intensity is quantified in Figure 6C, this cannot distinguish the change in cell number vs WFA expression. It is also unclear if GM6001 also affect PV or PV remain similar to results in Figure 1. Integrity analysis in Fig.6 e-h also remains qualitative. It is also essential to show if GM6001 impacts PNNs integrity "qualitatively", and also electrophysiological properties of excitatory and PV interneurons in Figure 7 & 9.

#5: Related to the above point, in Figure 5, the gelatinase activity between Sham and 400-600um are not significantly different, yet the WFA levels are significantly different. Going back to the concern regarding connection of PV cell loss to WFA loss, the authors should, if possible, consider adding PV labeling to normalize WFA values to PV cell loss, or open in discussion alternative mechanisms that may contribute to the loss of PNN. Even the use of a broad-spectrum MMP blocker like GM6001 can only partially recover PNN, even though gelatinase activity decreases a lot more by proportion even in the glioma. Again, could there be an additional or alternative mechanism that's different from proteolytic action of MMPs worth mentioning in discussion?

Minor Comments:

#6: The introduction contains very thorough description of the key players to the current work (enhanced glutamatergic drive by SXC, loss of GABAergic inhibition, PV and PNN, and MMP as major catalyst of PNNs). However the overall flow does not explicitly mention the gaps in the current knowledge, and as a reader it feels difficult to clearly map out what is exactly missing or unknown, and what exactly the authors aim to answer in their work. This may be in fact the reflection of a flaw in logic to relate ChABC parts to glioma parts giving an impression that these 2 parts may not fit well as a single manuscript which is a bigger issue I pointed out earlier.

#7: In Figure 1h, according to the significance labels only NeuN and PV comparisons for 0.2-0.4mm and 0.4-0.6mm seem to be significantly different, but the results in the main text states both WFA and PV were significantly more affected? Just from the figure it seems like NeuN and WFA for 0.2-0.4 and 0.4-0.6 should be significantly different?

#8: The distance units for all figures and in main text should be consistent (use only mm or only um).

#9: Figure 1g and 3d y-axis label should say "WFA" or "WFA(PNN)" instead of "PNN" for consistency with IHC representative images.

#10: The representative images in Figure 3a and b seem like most of the selected are well within 200um from the glioma border. According to bar graphs in this area there should not be any difference in PV or WFA between GBM22 and 14 in this area but the representative images seem to show clear differences. Perhaps it may be better to choose an image from 400~600um that better represents the difference according to distance?

#11: Figure 4 layout is a bit confusing and hard to follow mainly due to the placement of inset C. We suggest editing the layout to have inset C moved next to inset B. The spaces can be adjusted by making the image sizes consistent and changing the width and spacing of the bar graphs.

#12: Figure 6 title should be reworded.

#13: Some values in figure legend don't have units like Supplementary fig.4h miss 'pF'

#14: Page 20 correct company name is Narishige

Reviewer #3 (Remarks to the Author):

The manuscript presents a thorough investigation of the electrophysiological changes wrought by glioma. There are two major findings presented in this manuscript, with emphasis on the impact of glioma on fast spiking interneurons. In brief, mice injected with human glioblastomas (GBM22 xenograft) that secrete glutamate and matrix metalloproteinases show firing rate increases in excitatory neurons and decreases in fast-spiking interneurons; xenografts of a glioma that does not secrete glutamate (GBM14), but does secrete matrix metalloproteinases only impacts the fast-spiking interneurons. This leads to further investigations of the impact of these metalloproteinases on fast-spiking cells, resulting in the observation that degradation of perineuronal nets increases fast-spiking cell membrane capacitance, causing a reduction in firing rates. These conclusions are drawn largely from data obtained from electrophysiological recordings of neurons in cortex in acute brain slices derived from mice injected with human glioblastomas from two cell lines. However, the authors are quite thorough in their interrogation, providing data on enzymatic activity, perineuronal net degradation, and astrocytic activity.

Overall, I quite like the manuscript. To my knowledge, the role of perineuronal nets as a regulator of membrane capacitance is new. Typically, they are thought to stabilize synapses by "gluing" them in place - an explanation that always left me unsatisfied. Their regulation of capacitance is far more compelling and explains not only the loss of inhibition associated with glioma, but also the results from Maffei and Hensch on adult cortical plasticity. Disinhibition, rather than synaptic glue, is a better explanation for their findings.

My major complaint is with the figures. Reporting mean and standard error is no longer an acceptable way to convey data. At a minimum, boxplots that show quartiles or, better still, violin plots, is the way to go. Also, it would be good to state at the outset the brain region from which the recordings were made. The tumors were injected at roughly 1.5mm lateral and 1mm anterior of bregma, which is primary motor cortex. Recordings were made up to 1mm away, I believe.

Again, the message is fresh and the data are compelling, forcing a rethinking of what PNNs do. I suggest only minor revisions to the data display in the figures. Otherwise, a nice paper that should be well received.

Reviewer #4 (Remarks to the Author):

In the present work authors have investigated the mechanisms and consequences of tumor on cortical GABAergic neurons.

To address this issue, authors have used a plethora of cell biology techniques, including immunocytochemistry, epifluorescence, electrophysiological recordings in brain slices. They analyzed the effects of xenotransplants of different types of human derived glioma cells on PNN and electrophysiological properties of interneurons in mice.

They show the degradation of perineuronal nets (PNNs) around cortical fast spiking interneurons in peritumoral regions. The PNN degradation occurring in peritumoral regions, results in the increase of neuronal membrane capacitance that leads to a decrease of interneuron firing rate, an effect that can be mimicked by chondroitinase enzyme. The idea that PNN critically controls interneuron firing rate is also supported with computational modelling.

This is an important, clear and well-performed study which adds valuable information regarding

several relevant topics. First, it demonstrates a novel function of PPN in regulating firing activity of interneurons. Second, it provides novel mechanisms likely involved in epileptogenesis associated with brain tumors. Third, this study is highly relevant because the proposed novel mechanism may impact our knowledge of the mechanisms involved in other brain diseases accompanied by neuroinflammation and tissue reorganization.

The results are novel, interesting and convincing. Technically and methodologically, the manuscript is elegant and detailed, and the experimental design and the analysis performed are adequate. The conclusions drawn are appropriately supported by the results.

Therefore, I have no major concerns about present manuscript and I feel it has merit to be published. There are however some minor issues that authors may address to enhance the high quality of the manuscript.

Specific comments

1. Page 6, end of 3rd paragraph. Authors state: "We conclude that these cells are either dead or in the process of dying due to Glu released from the tumor cells". This is a likely hypothesis based on previous findings in the literature, as adequately described by authors, but no evidence is presented in this manuscript. This is a strong conclusion that needs to be experimentally supported, otherwise, the statement should be toned down as a suggestion.

2. Page 9, 3rd paragraph. Results presented in Fig. 7e and Fig. S4b,c need to be clarified. Authors state that the effects of GBM 22 were absent in GBM 14 tumors. However, graph presented in Fig. 7e shows intermediate effects between GBM 22 and sham. These results suggest that additional mechanisms are present. Alternatively, it is simply due to a the relatively small sample size (n=9 vs 27 and 36 cells). Increasing the sample size for GBM 14, would help to clarify the issue.

3. Related to previous point, authors hypothesized that the lower firing threshold and the depolarized membrane potential of principle neurons in PTC were due to chronically elevated Glu being released from the nearby tumor. To test the hypothesis, they used GBM14 tumors that do not release Glu. An additional and perhaps more straight forward experiment would be testing the effects of AMPA and NMDA receptor antagonists. If the authors' hypothesis is correct the observed changes in principal neurons would be abolished.

Reviewers' comments:

Reviewer #1

(Remarks to the Author):

This study by Tewari et al. builds on previous reports from their and other labs that brain tumours are associated with epilepsy and that this not only involves excess Glu release by tumor-related cells but possibly also loss of GABAergic interneurons (particularly fast-spiking neurons (FSNs) and their associated perineuronal nets (PNNs)). Here they address directly the FSN and PNN loss in mouse brain tumour models. They show that loss of PV interneurons and PNNs is particularly strong when epileptogenic gliomas are used in mice, and that the PNN loss is correlated with gelatinase activity and reduced by inhibitors of gelatinase. They then show that FSNs exhibit greatly reduced firing frequencies in the vicinity of gliomas (both epileptogenic and non-epileptogenic), and that disruption of PNNs in slices leads to reduced firing of PNNs and enhanced burst firing by excitatory neurons. The authors use modelling to suggest a mechanism in which PNNs have myelin-like properties, enhancing the excitability of FSNs.

This study provides interesting observations relating brain tumours, FSN dysfunction and PNN loss, and suggesting possible causal relationships between these processes and epileptogenesis induced by brain tumours. They further provide potentially important insights as to the role of PNNs in FSN and network function. However, the evidence relating causally PNN loss, FSN dysfunction and epileptogenesis induced by tumors is not compelling. Notably, there seems to be an important discrepancy between how non-epileptogenic gliomas have only a modest effect on PV and PNN loss compared to epileptogenic ones (Fig. 3), and the fact that both types of gliomas have strong and undistinguishable effects on FSN firing rates in situ. Furthermore, the study ignores a vast literature that has related degradation of PNNs to enhanced plasticity and brain repair, without detrimental effects to network activity in situ (including in settings such as stroke models, where seizures can be an issue).

Overall, this study reports on a set of interesting observations concerning FSNs and PNNs, but their significance for tumor-induced epilepsy and for PNN function is not entirely clear.

Specific points:

1). In the opinion of this Reviewer, one main issue concerns a comparison of the findings in Fig 3 (epileptogenic gliomas are much more effective at inducing PV and PNN loss than non-epileptogenic ones) and in Fig. 7 (undistinguishable effects of epileptogenic and non-epileptogenic gliomas on the (reduced) firing properties of FSNs in situ). These combined data suggest that reduced FSN firing is poorly correlated to PNN loss in situ, and not correlated to how gliomas induce epilepsy. This appears to contradict some of the main conclusions of the study.

Response:

We apologize having failed to more clearly state what these figures are showing. The recordings in Fig 7I (Fig 6I in revised manuscript) are also from FSNs with degraded PNNs although in non-epileptic GBM14 PTC. These tumors, when injected into cortex, showed PNN degradation but only in the immediate vicinity of the tumor, i.e., up to 0.2 mm from the tumor border, and the data in fig 7I (Fig 6I in revised manuscript) was obtained from neurons that had degraded PNNs. Therefore, the data shows that spike frequency of FSNs is equally altered in both glioma types provided PNNs are degraded regardless of their glutamate releasing property. The only difference between the two tumor types was that PNN degradation was much more widespread in GBM22 than GBM14 as shown in Fig 3. We have clarified the description of this figure in the revised manuscript on page 10 and added representative images of the different experimental groups including the recorded peritumoral FSN from GBM14-implanted cortex (See Supplementary Fig 2d in revised manuscript), which showed lower spike frequency and disintegrated PNN.

2) The key data relating gelatinase activity inhibition to PNN protection in situ are not as compelling as one would like them to be. Gelatinase inhibitors only have a modest effect on preventing PNN disassembly (Fig. 6). In these experiments, gelatinase activity was not completely inhibited, which might explain the modest protective effect on PNNs. However, since this is such a central point of the study it would be important to better relate the two phenomena: can gelatinase be inhibited more completely and does that then result in complete PNN protection?

And what about epileptic activity induced by the glioma: is it also dependent on gelatinase activity?

Response:

a. In the revised manuscript, we have more carefully analyzed the effect of the MMPs inhibitor GM6001 on PNNs using two parameters: PNN integrity reflected in the 3D architecture of PNNs, and the total numbers of PNNs in PTC. With these quantitative readouts, we were able to demonstrate that MMPs block by GM6001 yielded a significant increase in the overall number of PNNs with up to 3-fold increase near the tumor (Revised Fig 5e), and also significantly more intact architecture (revised Fig 5h). Therefore, the PNN degradation can be primarily attributed to the MMPs. In the revised manuscript, we merged Fig 5 and Fig 6 with new experiment data putting all relevant GM6001 data in one figure for improved clarity.

b. Regarding a complete rescue of PNNs with Gelatinase inhibitors, PNNs can be degraded by various MMPs, ADAMTs, TIMPS, and Hyase that are released from various CNS cells including glioma cells. Therefore a complete rescue of PNNs would require specific inhibitors for all the above enzymes and these do not currently exist. Note that we do not imply that the degradation of PNNs is exclusively due to MMPs, however, our data strongly suggests that they play a major role in PNN degradation in PTC.

c. We have adduced several papers in the discussion (highlighted text page 16) that show a substantial inhibition of seizure activity on treatment with MMPs inhibitors in other epilepsy models (Pollock, Everest et al. 2014, Dubey, McRae et al. 2017). One study suggest that glutamate released during seizures elevates MMPs activity levels (Rempe, Hartz et al. 2018). The challenge to studying specific effects of MMP inhibitors in epilepsy associated with glioma is the pleotropic effects of MMPs in tumor invasion, angiogenesis and growth (Rao 2003). Moreover, extracellular glutamate released from the tumor appears to be a major driver of epilepsy (Buckingham, Campbell et al. 2011, Campbell, Robel et al. 2015, Robert, Buckingham et al. 2015), therefore, complete rescue of seizures with MMPs inhibition is probably difficult to achieve.

3) The authors equate PV signal loss to loss of PV neurons. In other settings, particularly schizophrenia, this has been shown not to be accurate: there can be a great reduction/loss of PV signals without loss of PV neurons. The authors find that PNN and PV neurons are affected at a greater distance from the tumor than are NeuN+ total neuron numbers. Possibly, the distant effects might reflect loss of PNN and PV protein without neuronal loss, which might be interesting.

Response:

It is a possibility that PV signal loss can be mistaken for the cell loss, however, the following findings suggest that not to be the case here:

a. In Fig 1, we used three markers to identify neurons: 1) WFA which labels PNN that is a fairly static structure, 2) PV which is a cytosolic marker that may be more futile, and 3) NeuN which is a stable non-specific neuronal nuclei marker. The density of PV+ versus WFA+ neurons was essentially identical for each binned distance analyzed and changed in lockstep, yet the density of all neurons, identified by NeuN was significantly different. If PV expression had been selectively downregulated, there should have been a discrepancy between the WFA and PV data.

b. We observed PV neurons at different stages of disintegration of their PNNs and added a figure in the revised manuscript with few representative images (supplementary Fig 1b-i) showing no apparent change in the PV expression.

4) The authors ignore a large number of studies that have related local PNN degradation in the adult brain to highly beneficial effects on brain plasticity and repair (and on sensitive periods for plasticity). How might those findings relate to their results here?

Response:

a. We are aware of the studies showing PNN degradation and synaptic plasticity and cited some of the seminal findings in the introduction. Yet these studies have not examined the biophysical properties of individual neurons and how PNNs may contribute to hyperexcitability. Nevertheless, changes in synaptic circuits and hence networks are known to contribute to epileptogenesis. We added this possibility to our discussion with appropriate citations of other works.

5) The discussion is largely devoted to speculations regarding how organised ECM in PNN might affect excitability, and how PNN loss might relate to epilepsy. What is missing, however, is a critical treatment of the findings in the paper, and of the extent to which the key conclusions are based on compelling evidence.

Response:

The primary focus of this paper was to report on a novel function of PNNs as it was discovered by accident in tumor-associated epilepsy where the endogenous release of MMPs destroys PNNs. We believe quite strongly that our data make a strong case that loss of PNNs alters the cells intrinsic excitability, and we show that this is likely due to MMP released from glioma and can be phenocopied by experimental real time digestion of PNNs absent a tumor. Structural organization of PNNs was emphasized throughout, as it was only in neurons with intact organized PNNs that the biophysical properties were altered upon PNN disintegration, not in excitatory neurons embedded in ECM. From this derives our suggestion that a complete surface coating of the membrane acts as an insulator that reduces cell capacitance.

Reviewer #2

(Remarks to the Author):

The current manuscript by Tewari et al. aims to describe novel mechanisms associated with the pathophysiology of tumor-associated epileptogenesis. Here, the authors provide evidences (but see major comments) specific to epileptogenic glioma GBM22 that support the two components of loss of GABAergic inhibition required for peritumoral epilepsy: 1) loss of PV interneurons due to excitotoxic levels of glu released by glioma, and 2) electrophysiological dysfunction of PV interneurons and loss of perineuronal nets partly due to glioma-secreted MMPs. Interestingly, the authors show that PNN degradation alone is enough to significantly reduce the firing rate of PV interneurons, enough to disrupt GABAergic tone. While some of the findings are interesting, there are major issues with this study. Most importantly, Glioma model data and ChABC model are not logically well connected (see major comment 2) giving an impression that two independent studies (Glioma and ChABC) are presented together with weak logical connections at least in this current form. In addition, experimental data/analysis is very weak to support Glu-independent/MMP-dependent impact to PNNs integrity (see major comment 1). Given that MMP is already known to mediate glioma induced seizure, it is also essential to more thoroughly link MMP and PNNs properties (see major comment 5).

Major Comments:

#1: One of the major points of this study is the MMP dependent (Glu independent) impact of Glioma to PNNs (independent to cell loss). However, there is no quantitative data supporting this point. While authors quantified WFA intensity, this readout does not distinguish PV cell loss vs PNN loss without cell death. While authors show qualitative representative images of PNNs integrity, there is no quantification. Without through quantification of PNNs integrity and analysis taking into account for the PV cell loss (double staining of WFA and PV), the major point of this manuscript is not supported.

Specifically, there seems to be no detailed methodological explanation for the process of line intensity profiling of cells in the experiments for Fig 2b-i and Fig 6 d-h. Moreover, there are no statistical comparisons involved that would show significance to the authors' claims. Specifically, in terms of Fig 2, "few viable neurons in far peritumoral zone..." – how many (#/area), and what is the exact distance? "Majority of neurons surrounded by invading glioma cells..." – how much (#/area), how many glioma cells = "surrounded"? Similarly, for Fig 6, the number of neurons observed is also not indicated. For consistency, the representative images in 6d-g should also include DAPI and gelatinase.

Response:

- a. As requested we have quantified the PNN integrity and PNN numbers in PTC, sham and GM6001 treated animals and the resulting data including statistical analysis is incorporated in revised Figs 2 and 5. Moreover, the detailed methodology for line intensity profiling is included in the revised manuscript.
- b. We have added a new heading in the method section with elaborate account of PNN integrity quantification and line profile methodology.
- c. We have added DAPI and gelatinase staining also in Fig 6 (Fig 5f in revised manuscript) as suggested.

#2: Another major claim of this study is that FSI firing frequency change in glioma model is due to changes in capacitance because of PNNs degradation. However, this is not supported by the results. Firing frequency in GBM22 and GBM14 groups changed equally but glioma GBM14 and glioma GBM22 affected capacitance differently (fig 7i vs supplemental fig 4f). Capacitance was not impacted by GBM14 suggesting that capacitance difference in GBM22 is likely the consequence of chronic exposure to glu from glioma. Given this dissociation between glioma model and ChABC model, there is a logical flaw in presenting glioma related data and ChABC data together in the same manuscript. Authors also failed to cite published work showed ChABC decrease FSI excitability (Balmer eNeuro 2016).

Responses:

- a. The pooled data included in our original manuscript Fig 7i and supplemental Fig 4f respectively indeed showed a significantly higher capacitance in neurons next to GBM22 but only a trend towards higher capacitance in GBM14 that was insignificant. As we suspected that the lack of significance may be due to the sample size, we elected to spend the past 3 months repeating these experiments with a larger sample size as this was also suggested by Reviewer#4 (Specific comment 2) and incorporated new data in its original version. Upon repeating these experiments in 21 cells from 8 mice, the GBM14-associated neurons showed the same statistically significant higher capacitance as observed in GBM22 hopefully alleviating this concern (see Fig 6l and supplementary Fig 4h in revised manuscript).
- b. We felt it was imperative to measure changes in capacitance in real time while recording from the cell. The use of ChABC to experimentally disrupt PNNs has become standard in the literature for this purpose, and has been used in studies of seizures and epilepsy in rodent models (Pollock, Everest et al. 2014, Dubey, McRae et al. 2017). ChABC is acting quickly and completely disintegrates PNNs within 60 minutes, making it an ideal tool for this purpose. By comparison MMPs are much slower acting. We are excited that the use of ChABC was able to show that degradation of PNNs is sufficient to increase the capacitance thereby decreasing AP frequency in the identical manner that the presence of the tumor disintegrates PNNs and increases capacitance.
- c. As suggested we included Balmer et al. 2016 in the revised discussion.

#3: It is not clear whether the firing frequency (fig 7e, 7l) are measured with or without synaptic blockers. As the amount of both excitatory and inhibitory cells change near glioma, it is obvious to expect that the number of synaptic contacts and synaptic drive changed. To support authors claim that changes shown in 7e (for pyramidal neurons) are extrinsic and 7l (for FS interneurons) are intrinsic, recording with synaptic blockers are necessary to support the authors' interpretation.

Response:

We did not use synaptic blockers in the original manuscript as our previous papers clearly showed that spike frequency alterations in excitatory neurons is primarily attributed to the glioma-released glutamate (Buckingham et al 2011, Campbell et al 2015, Robert et al 2015), and, consistent with that, excitatory pyramidal neurons in the PTC of non-glutamate releasing GBM14 did not show any significant difference in the spike frequency than the sham. However, we like to assure the reviewer through using APV and CNQX as proposed (see figure below). This figure is not currently part of the manuscript but could be added to supplementary figures if so desired (n=27 (GBM22), n=36 (Sham), n= 8 (GBM22 (APV+CNQX)).

#4: It would be very important to thoroughly assess the “causal” contribution of MMP to the properties of PNNs. While WFA intensity is quantified in Fig 6C, this cannot distinguish the change in cell number vs WFA expression. It is also unclear if GM6001 also affect PV or PV remain similar to results in Fig 1. Integrity analysis in Fig.6 e-h also remains qualitative. It is also essential to show if GM6001 impacts PNNs integrity “qualitatively”, and also electrophysiological properties of excitatory and PV interneurons in Fig 7 & 9.

Response:

We performed new experiments and analyzed PNN counts as well as integrity in the different experimental groups and included this data in the revised manuscript which further substantiate our conclusions (see new Fig 5e-h in the revised manuscript).

We also performed additional experiments to study the effect of GM6001 on the electrophysiological properties of excitatory and PV interneurons in the GBM22 peritumoral cortex. Electrophysiological properties of FSNs in the peritumoral cortex of GM6001-treated mice showed significant recovery of spike frequency and membrane capacitance and data is presented in Fig 7i and 7j, respectively. Excitatory neurons in the GBM 22 PTC of GM6001-treated mice exhibited intermediate results suggesting an outcome of interplay of elevated glutamate and intact PNNs (Fig 7e revised manuscript). None of the passive properties of excitatory neurons differ in the GM6001-treated group. We have added representative images of recorded FSNs in different groups including GBM22 PTC with GM6001 treatment which show remarkably intact PNNs compared to that from non-treated glioma implanted brains (Supplementary Fig 2c-f).

#5: Related to the above point, in Fig 5, the gelatinase activity between Sham and 400-600um are not significantly different, yet the WFA levels are significantly different. Going back to the concern regarding connection of PV cell loss to WFA loss, the authors should, if possible, consider adding PV labeling to normalize WFA values to PV cell loss, or open in discussion alternative mechanisms that may contribute to the loss of PNN. Even the use of a broad-spectrum MMP blocker like GM6001 can only

partially recover PNN, even though gelatinase activity decreases a lot more by proportion even in the glioma. Again, could there be an additional or alternative mechanism that's different from proteolytic action of MMPs worth mentioning in discussion?

Response:

There might not be an absolute spatial correlation between gelatinase activity and PNN degradation (WFA intensity) due to the fact that gelatinase assay reflects only MMP2 and MMP9 activity but PNN degradation can also be achieved by other MMPs and factors released from glioma and reactive glial cells as discussed below. Therefore we completely agree with the point that there might be additional mechanisms to glioma-mediated degradation of PNNs. For example glioma cells and reactive glial cells are the primary source of wide range of extracellular remodeling molecules including MMPs, ADAMTs, ADAMs, Hyase, TIMPs, etc (Rao 2003, Kim, Porter et al. 2016). However, we mainly focused on the MMPs due to the fact that high-grade glioma release MMPs to facilitate invasion (Rao 2003) and we observed a gradient of PNN degradation from the glioma border suggesting that glioma could be the source of PNN degrading agents (Fig 5b revised). In addition, the most common sources of PNN remodeling agents, reactive glial cells did not show a spatial correlation with the PNN degradation (Fig 4 and supplementary Fig 3b-c). Glioma cells are also known to release ADAMTs that have proteoglycanase activity and can potentially degrade PNNs. Hyase is another potential candidate for PNN degradation released from glioma (Stern 2008). These details are already present in the beginning of result 2 and we have added the possibility of additional mechanisms in the discussion section of the revised version.

Minor Comments:

#6: The introduction contains very thorough description of the key players to the current work (enhanced glutamatergic drive by SXC, loss of GABAergic inhibition, PV and PNN, and MMP as major catalyst of PNNs). However the overall flow does not explicitly mention the gaps in the current knowledge, and as a reader it feels difficult to clearly map out what is exactly missing or unknown, and what exactly the authors aim to answer in their work. This may be in fact the reflection of a flaw in logic to relate ChABC parts to glioma parts giving an impression that these 2 parts may not fit well as a single manuscript which is a bigger issue I pointed out earlier.

Response:

We have added necessary content to make the questions of this study more explicit. We have already made our points in the response of major comment 2 to clarify the use of ChABC to mimic the PNN degradation by glioma-released MMPs.

#7: In Fig 1h, according to the significance labels only NeuN and PV comparisons for 0.2-0.4mm and 0.4-0.6mm seem to be significantly different, but the results in the main text states both WFA and PV were significantly more affected. Just from the figure it seems like NeuN and WFA for 0.2-0.4 and 0.4-0.6 should be significantly different?

Response:

We have added the significant labels and updated the legend of Fig.1h with the statistical details accordingly.

#8: The distance units for all figures and in main text should be consistent (use only mm or only um).

Response:

We have changed all labels to mm in the revised version.

#9: Fig 1g and 3d y-axis label should say "WFA" or "WFA(PNN)" instead of "PNN" for consistency with IHC representative images.

Response:

We have made the appropriate changes in the revised version.

#10: The representative images in Fig 3a and b seem like most of the selected are well within 200um from the glioma border. According to bar graphs in this area there should not be any difference in PV or WFA between GBM22 and 14 in this area but the representative images seem to show clear differences. Perhaps it may be better to choose an image from 400~600um that better represents the difference according to distance?

Response:

We have replaced the previous image with new images showing a bigger field of view in the revised version.

#11: Fig 4 layout is a bit confusing and hard to follow mainly due to the placement of inset C. We suggest editing the layout to have inset C moved next to inset B. The spaces can be adjusted by making the image sizes consistent and changing the width and spacing of the bar graphs.

Response:

As reviewer pointed out, the layout was for space adjustment and we have changed the layout accordingly in the revised version.

#12: Fig 6 title should be reworded.

Response:

There was a typographic error and has been corrected in the revised version. Fig 5 and 6 of the original manuscript were combined and presented as Fig 5 in the revised version.

#13: Some values in figure legend don't have units like Supplementary fig.4h miss 'pF'

Response:

We have corrected it in revised version.

#14: Page 20 correct company name is Narishige.

Response:

We have corrected it in revised version.

Reviewer #3

(Remarks to the Author):

The manuscript presents a thorough investigation of the electrophysiological changes wrought by glioma. There are two major findings presented in this manuscript, with emphasis on the impact of glioma on fast spiking interneurons. In brief, mice injected with human glioblastomas (GBM22 xenograft) that secrete glutamate and matrix metalloproteinases show firing rate increases in excitatory neurons and decreases in fast-spiking interneurons; xenografts of a glioma that does not secrete glutamate (GBM14), but does secrete matrix metalloproteinases only impacts the fast-spiking interneurons. This leads to further investigations of the impact of these metalloproteinases on fast-spiking cells, resulting in the observation that degradation of perineuronal nets increases fast-spiking cell membrane capacitance, causing a reduction in firing rates. These conclusions are drawn largely from data obtained from electrophysiological recordings of neurons in cortex in acute brain slices derived from mice injected with human glioblastomas from two cell lines. However, the authors are quite thorough in their interrogation, providing data on enzymatic activity, perineuronal net degradation, and astrocytic activity.

Overall, I quite like the manuscript. To my knowledge, the role of perineuronal nets as a regulator of membrane capacitance is new. Typically, they are thought to stabilize synapses by "gluing" them in place - an explanation that always left me unsatisfied. Their regulation of capacitance is far more compelling and explains not only the loss of inhibition associated with glioma, but also the results from Maffei and Hensch on adult cortical plasticity. Disinhibition, rather than synaptic glue, is a better explanation for their findings.

My major complaint is with the figures. Reporting mean and standard error is no longer an acceptable way to convey data. At a minimum, boxplots that show quartiles or, better still, violin plots, is the way to go. Also, it would be good to state at the outset the brain region from which the recordings were made. The tumors were injected at roughly 1.5mm lateral and 1mm anterior of bregma, which is primary motor cortex. Recordings were made up to 1mm away, I believe.

Again, the message is fresh and the data are compelling, forcing a rethinking of what PNNs do. I suggest only minor revisions to the data display in the figures. Otherwise, a nice paper that should be well received.

Response:

We appreciate the comments and have replaced the bar diagrams with box and whisker plots in revised version. We have also modified sentences and added the details in method section too.

Precise determination of recording location as motor cortex in our experiments would be difficult to mention because growth of glioma varies from animal to animal and recorded cortical areas were both in motor and somatosensory cortex. However, in the method section of revised manuscript we have mentioned motor and somatosensory cortices as areas from where recordings were made.

Reviewer #4
(Remarks to the Author):

In the present work authors have investigated the mechanisms and consequences of tumor on cortical GABAergic neurons. To address this issue, authors have used a plethora of cell biology techniques, including immunocytochemistry, epifluorescence, electrophysiological recordings in brain slices. They analyzed the effects of xenotransplants of different types of human derived glioma cells on PNN and electrophysiological properties of interneurons in mice. They show the degradation of perineuronal nets (PNNs) around cortical fast spiking interneurons in peritumoral regions. The PNN degradation occurring in peritumoral regions, results in the increase of neuronal membrane capacitance that leads to a decrease of interneuron firing rate, an effect that can be mimicked by chondroitinase enzyme. The idea that PNN critically controls interneuron firing rate is also supported with computational modelling.

This is an important, clear and well-performed study which adds valuable information regarding several relevant topics. First, it demonstrates a novel function of PPN in regulating firing activity of interneurons. Second, it provides novel mechanisms likely involved in epileptogenesis associated with brain tumors. Third, this study is highly relevant because the proposed novel mechanism may impact our knowledge of the mechanisms involved in other brain diseases accompanied by neuroinflammation and tissue reorganization.

The results are novel, interesting and convincing. Technically and methodologically, the manuscript is elegant and detailed, and the experimental design and the analysis performed are adequate. The conclusions drawn are appropriately supported by the results.

Therefore, I have no major concerns about present manuscript and I feel it has merit to be published. There are however some minor issues that authors my address to enhance the high quality of the manuscript.

Specific comments:

1. Page 6, end of 3rd paragraph. Authors state: “We conclude that these cells are either dead or in the process of dying due to Glu released from the tumor cells”. This is a likely hypothesis based on previous findings in the literature, as adequately described by authors, but no evidence is presented in this manuscript. This is a strong conclusion that needs to be experimentally supported, otherwise, the statement should be toned down as a suggestion.

Response:

We have modified the statement accordingly in revised version, however the corresponding Fig 2b-h of the original manuscript are now in supplementary Fig 1b-i of the revised manuscript. This change was made to address the Reviewer#2's concern.

2. Page 9, 3rd paragraph. Results presented in Fig. 7e and Fig. S4b,c need to be clarified. Authors state that the effects of GBM 22 were absent in GBM 14 tumors. However, graph presented in Fig.7e shows intermediate effects between GBM 22 and sham. These results suggest that additional mechanisms are present. Alternatively, it is simply due to the relatively small sample size (n=9 vs 27 and 36 cells). Increasing the sample size for GBM 14, would help to clarify the issue.

Response:

In Fig 7e of the original manuscript, the effects appeared intermediate but not statistically significant. In the revised manuscript we increased sample size as suggested and current data explicitly substantiate our conclusion of absence of enhanced spike frequency of excitatory neurons in the non-glutamate

releasing tumors GBM14 (Fig 6e in the revised manuscript). Including this additional data from GBM14 in other relevant figures (e.g., Supplementary Fig 4h) further substantiate our interpretations and we thank reviewer for this valuable suggestion.

3. Related to previous point, authors hypothesized that the lower firing threshold and the depolarized membrane potential of principle neurons in PTC were due to chronically elevated Glu being released from the nearby tumor. To test the hypothesis, they used GBM14 tumors that do not release Glu. An additional and perhaps more straight forward experiment would be testing the effects of AMPA and NMDA receptor antagonists. If the authors' hypothesis is correct the observed changes in principal neurons would be abolished.

Response:

Please see reviewer #2, point 5, where we addressed this very issue and included a data figure that may or may not be important to include as supplementary data.

References:

Buckingham, S. C., et al. (2011). "Glutamate release by primary brain tumors induces epileptic activity." Nature medicine **17**(10): 1269-1274.

Campbell, S. L., et al. (2015). "GABAergic disinhibition and impaired KCC2 cotransporter activity underlie tumor-associated epilepsy." Glia **63**(1): 23-36.

Dubey, D., et al. (2017). "Increased metalloproteinase activity in the hippocampus following status epilepticus." Epilepsy Research **132**: 50-58.

Kim, S. Y., et al. (2016). "A potential role for glia-derived extracellular matrix remodeling in postinjury epilepsy." Journal of neuroscience research **94**(9): 794-803.

Pollock, E., et al. (2014). "Metalloproteinase inhibition prevents inhibitory synapse reorganization and seizure genesis." Neurobiology of disease **70**: 21-31.

Rao, J. S. (2003). "Molecular mechanisms of glioma invasiveness: the role of proteases." Nature Reviews Cancer **3**: 489.

Rempe, R. G., et al. (2018). "Matrix Metalloproteinase-Mediated Blood-Brain Barrier Dysfunction in Epilepsy." The Journal of Neuroscience **38**(18): 4301-4315.

Robert, S. M., et al. (2015). "SLC7A11 expression is associated with seizures and predicts poor survival in patients with malignant glioma." Science translational medicine **7**(289): 289ra286-289ra286.

Stern, R. (2008). Hyaluronidases in cancer biology. Seminars in cancer biology, Academic Press.

Reviewers' comments:

Reviewer #1 (Remarks to the Author):

The Authors have carefully addressed the points raised by the Reviewers.

This study provides a novel perspective on the function of PNNs (and indirectly also on the function of PV neurons) - this constitutes an important advance.

Reviewer #2 (Remarks to the Author):

In this revised manuscript, authors addressed one of major concerns in the original manuscript (discrepancy between FSI firing and capacitance in GBM22 and 14 models) by adding more animals. They also newly examined the effect of MMP inhibitors to FS properties. However, there are several key issues remain unaddressed.

#1: This revised manuscript did not include key quantification of PV/PNN co-localization (also questioned by rev1) failing to address the major concern on distinguishing PNN-cell loss vs PNN or PV loss without cell death. Given that the phenotype is a mixture of PNN cell loss and PNN integrity for the remaining cell, it is essential to distinguish the two by quantifying co-localization analysis of PNN/PV (and NeuN). Given that WFA+cell density is higher than PV+cell density some case, more detailed quantification is necessary.

#2: Related to the above point, WFA intensity data in Fig2b, 3ef, 5Ci is not very informative because the intensity change can reflect either cell death or PNN integrity deficits in remaining cells. Authors should also present this data normalized by WFA-cell density. In addition, PNN integrity analysis (fig2e, 5h) should be more thoroughly conducted as a function of distance from glioma.

3: While it is critical for this study to show FS interneuron firing property is intrinsic, authors failed to present the data with synaptic blockers for FS interneurons (only data with pyramidal cells were presented in rebuttal).

4: New data in Fig.5e indicates that MMP activity not only impacts PNN integrity but surprisingly also the WFA+cell survival. This indicates a rather complicated role of both glutamate and MMP in cell survival. This additional role of MMP on survival needs to be explicitly stated and discussed in the text throughout.

Reviewer #4 (Remarks to the Author):

Authors have adequately addressed the concerns about the previous version.

I have no further comments.

Reviewers' comments:

Reviewer #1 (Remarks to the Author):

The Authors have carefully addressed the points raised by the Reviewers.

This study provides a novel perspective on the function of PNNs (and indirectly also on the function of PV neurons) - this constitutes an important advance.

Response: We thank the reviewer for thorough review and recognizing the novelty and importance of our current study.

Reviewer #2 (Remarks to the Author):

In this revised manuscript, authors addressed one of major concerns in the original manuscript (discrepancy between FSI firing and capacitance in GBM22 and 14 models) by adding more animals. They also newly examined the effect of MMP inhibitors to FS properties. However, there are several key issues remain unaddressed.

#1: This revised manuscript did not include key quantification of PV/PNN co-localization (also questioned by rev1) failing to address the major concern on distinguishing PNN-cell loss vs PNN or PV loss without cell death. Given that the phenotype is a mixture of PNN cell loss and PNN integrity for the remaining cell, it is essential to distinguish the two by quantifying co-localization analysis of PNN/PV (and NeuN). Given that WFA+cell density is higher than PV+cell density some case, more detailed quantification is necessary.

Responses:

a. We have now quantified the PV/PNN co-localization in the peritumoral cortex as suggested and the resultant data confirms that PV cell loss was not mistaken for a decrease in PV expression. We says this for the following reasons:

1. If PV were selectively downregulated in the peritumoral cortex and consequently mistaken for cell loss then the number of neurons that are NeuN⁺+WFA⁺+PV⁻ should have increased in PTC compared to control. However, our quantification shows no significant difference in this phenotype of staining in any of the PTC spatial bins as well as controls (Please see Supplementary Fig. 1c (left)).
2. Furthermore, the ratio of PV⁺/WFA⁺ cells should be different among PTC spatial bins (decrease if PV is downregulated). Note that our co-localization data also shows that the ratio of PV⁺/WFA⁺ cells are not significantly different among PTC bins and control (See Supplementary Fig. 1c (right)).
3. Peritumoral cortex should show a gradient of lower to higher (or normal) PV expression in the peritumoral cortex, however, no such gradient appears in any of the peritumoral images (Fig. 1a, Supplementary Fig1e).

b. We would like to further clarify that our data in Fig. 1h which is now Fig 1e (after 2nd revision) shows that the density of WFA⁺ cell is not statistically significantly different than PV⁺ cell in any spatial bin (See statistics in supplementary table 1).

#2: Related to the above point, WFA intensity data in Fig2b, 3ef, 5Ci is not very informative because the intensity change can reflect either cell death or PNN integrity deficits in remaining cells. Authors should also present this data normalized by WFA-cell density. In addition, PNN integrity analysis (fig2e, 5h) should be more thoroughly conducted as a function of distance from glioma.

Responses:

a. Using published methods for PNN analysis, we have conducted a thorough PNN analysis by measuring WFA intensity as a function of distance from tumor.

Regarding the normalization of WFA intensity to the cell count, we are confident that our approach for WFA intensity quantification ruled out the possibility of having low WFA intensity in PTC groups due to less number of PNNs/cells. WFA intensity was calculated by placing uniform sized ROIs onto individual PNNs instead of taking average intensity of whole image or a large ROI with different number of PNNs. Therefore, WFA intensity in all the data points represents mean WFA intensity of a single PNN in different experimental groups. This rules out the possibility of lower WFA intensity to be mistaken for fewer PNNs/cells in PTC. We have more clearly articulated this detail in the method section under “Quantification of IHC and ISZ images” subheading (Manuscript Page 20).

b. We presented data for WFA cell counts in all the mentioned figures except 3e,f. Data presented in the latter did not seek to study cell death in the CA2 area where PNNs are present around the excitatory neurons, but instead were aimed to show a canonical CSPG degradation by glioma. This objective is now explicitly mentioned in the results section (see highlighted text in manuscript Page 6).

c. Quantifying PNN architecture by counting high intensity WFA peaks was done in the 0-0.4mm PTC area to provide an exemplary supplementary evidence (similar to Fig 4b and 4c where GFAP and WFA intensity was quantified in only 2 spatial bins) based on the fact that 0-0.4mm PTC represents the most affected area from glioma. In addition, the spatial bins within this area (0-0.2mm and 0.2-0.4mm) are not significantly different in WFA intensities (e.g., Fig 2b and Fig 5c), therefore breaking down it to smaller bins provides no additional information.

3: While it is critical for this study to show FS interneuron firing property is intrinsic, authors failed to present the data with synaptic blockers for FS interneurons (only data with pyramidal cells were presented in rebuttal).

Response:

As suggested, in the revised manuscript, we have added data from new experiments on pyramidal cells as well as FSNs (See Supplementary Fig 5b,d), which substantiates the notion that altered spiking of FSNs is intrinsic whereas that of pyramidal neurons is extrinsic and glutamate driven. We have also updated the manuscript (Please see highlighted text in manuscript page 8).

4: New data in Fig.5e indicates that MMP activity not only impacts PNN integrity but surprisingly also the WFA+cell survival. This indicates a rather complicated role of both glutamate and MMP in cell survival. This additional role of MMP on survival needs to be explicitly stated and discussed in the text throughout.

Response:

We agree that glutamate and MMPs have a multitude of targets and therefore their inhibition can have unexpected outcomes (Rempe, Hartz et al. 2018). However, preservation of PNNs is one of the expected outcomes based on the well documented cleavage of PNN components by MMPs. Preservation of PNNs may in turn enhance cell survival as PNNs are known to play neuroprotective roles (Cabungcal, Steullet et al. 2013). In the revised manuscript, we have discussed these possibilities in both the results (Highlighted text in manuscript page 7) and discussion (Highlighted text in manuscript page 13) section as requested.

Cabungcal, J.-H., et al. (2013). "Perineuronal nets protect fast-spiking interneurons against oxidative stress." Proceedings of the National Academy of Sciences **110**(22): 9130-9135.

Rempe, R. G., et al. (2018). "Matrix Metalloproteinase-Mediated Blood-Brain Barrier Dysfunction in Epilepsy." The Journal of Neuroscience **38**(18): 4301-4315.

Reviewer #4 (Remarks to the Author):

Authors have adequately addressed the concerns about the previous version.
I have no further comments.

Response: We thank reviewer for thorough revision and insightful comments, which helped us to substantially improve our manuscript.

REVIEWERS' COMMENTS:

Reviewer #2 (Remarks to the Author):

Authors addressed the remaining concerns I had in the previous manuscript. I have no further comments.